# A pH-sensitive closed-loop nanomachine to control hyperexcitability at the single neuron level

Assunta Merolla[1,2,6], Caterina Michetti [1,3,6], Matteo Moschetta[1,2], Francesca Vacca[1], Lorenzo Ciano[1,3], Laura Emionite[2], Simonetta Astigiano [2], Alessandra Romei[1], Simone Horenkamp [1], Ken Berglund [4], Robert E. Gross[4], Fabrizia Cesca[1,5] ✉, Elisabetta Colombo [1,2] ✉ & Fabio Benfenati [1,2]

Epilepsy affects 1% of the general population and 30% of patients are resistant to antiepileptic drugs. Although optogenetics is an efficient antiepileptic strategy, the difficulty of illuminating deep brain areas poses translational challenges. Thus, the search of alternative light sources is strongly needed. Here, we develop pH-sensitive inhibitory luminopsin (pHIL), a closed-loop chemo-optogenetic nanomachine composed of a luciferase-based light generator, a fluorescent sensor of intracellular pH ($E^2GFP$), and an optogenetic actuator (halorhodopsin) for silencing neuronal activity. Stimulated by coelenterazine, pHIL experiences bioluminescence resonance energy transfer between luciferase and $E^2GFP$ which, under conditions of acidic pH, activates halorhodopsin. In primary neurons, pHIL senses the intracellular pH drop associated with hyperactivity and optogenetically aborts paroxysmal activity elicited by the administration of convulsants. The expression of pHIL in hippocampal pyramidal neurons is effective in decreasing duration and increasing latency of pilocarpine-induced tonic-clonic seizures upon in vivo coelenterazine administration, without affecting higher brain functions. The same treatment is effective in markedly decreasing seizure manifestations in a murine model of genetic epilepsy. The results indicate that pHIL represents a potentially promising closed-loop chemo-optogenetic strategy to treat drug-refractory epilepsy.

Epilepsy is a frequent disease affecting about 1% of the general population and characterized by recurrent seizures as a common denominator. Notwithstanding the success of anti-epileptic drugs, nearly one-third of the patients does not respond to the available therapies[1,2]. Many drug-resistant epilepsies have a local origin, being triggered by the presence of a *focus* of hyperexcitable neurons. In focal cortical dysplasia, temporal lobe epilepsy, traumatic brain injury, stroke, and some developmental encephalopathies, neurons in the epileptic *focus* experience an excitation/inhibition imbalance that puts them in a metastable state and can generate paroxysmal activity, eventually propagating to the surrounding healthy tissue[3]. In addition to the surgical removal of the epileptic focus, if it does not involve eloquent brain areas, the possibility to target the *focus* neurons with gene therapy and turn them off represents an effective strategy to prevent seizures[4].

[1]Center for Synaptic Neuroscience and Technology, Istituto Italiano di Tecnologia, Genova, Italy. [2]IRCCS Ospedale Policlinico San Martino, Genova, Italy. [3]Department of Experimental Medicine, University of Genova, Genova, Italy. [4]Department of Neurosurgery, Emory University School of Medicine, Atlanta, GA, USA. [5]Department of Life Sciences, University of Trieste, Trieste, Italy. [6]These authors contributed equally: Assunta Merolla, Caterina Michetti. ✉e-mail: fabrizia.cesca@iit.it; elisabetta.colombo@iit.it

Several groups successfully tried to abort drug-resistant paroxysmal activities in experimental models of epilepsy using a variety of gene therapy strategies, from CRISPR/dCas9 transcriptional stimulation of healthy alleles in haploinsufficiency to the overexpression of hyperpolarizing K[+] channels or inhibitory neuropeptides modulating the expression of neurotransmitters or ion channels[4–8]. Several studies have shown that optogenetics is a powerful therapeutic approach to control seizures[9,10]. Since the first demonstrations using *Natromonas* halorhodopsin (NpHR) to induce light-evoked inhibition of the paroxysmal activity of principal excitatory neurons or channelrhodopsin to optogenetically activate inhibitory interneurons in hippocampal slices[11,12], optogenetics has been successfully translated in-vivo. An enhanced form of NpHR (eNpHR3.0) was successfully targeted in-vivo to excitatory neurons of the hippocampus through the CaMKIIα promoter[13] and was shown to delay or abort seizure activity and decrease seizure duration[14–16]. Alternatively, parvalbumin promoter-driven channelrhodopsin2 expression in subpopulations of inhibitory interneurons in the hippocampus, ipsilateral or contralateral to the epileptic *focus*, was also effective in aborting or shortening epileptic seizures[14,17]. Although red-shifted microbial opsins have been engineered, thus far, the optogenetic strategy for seizure inhibition requires bringing light to the focus through chronically implanted optical fibers due to the poor tissue penetration of visible light.

To eliminate the burden of chronic optical fiber implants and local increases in temperature caused by high-intensity light stimuli[18,19], endogenous light generation at the site of genetic transduction has been developed by fusing the opsin actuator with luciferase variants in a chimera, promoting the activation of the opsin "*on demand*" in a cell-autonomous fashion by the systemic administration of a luciferase substrate. Based on this chemo-optogenetic strategy, a variety of luminescent opsins or "*luminopsins*", both excitatory and inhibitory, have been successfully engineered, in which *Gaussia* luciferase was fused with channelrhodopsin2 and archeorhodopsin, respectively[20]. More recently, a class of inhibitory luminopsins (iLMO) has been engineered, in which NpHR is coupled to the efficient light-generating chimera Nanolantern[21,22] composed of enhanced *Renilla* luciferase (Rluc) and the green fluorescent protein (GFP) variant Venus with high bioluminescence resonance energy transfer (BRET) efficiency[9,23–25]. Chemically activated iLMOs were shown to suppress action potential (AP) firing and synchronous bursting both ex-vivo and in-vivo[23,26].

Both optogenetic and chemo-optogenetic approaches do not solve the problem of the timing of either external or internal light stimulation with respect to incoming seizures. To this aim, closed-loop systems have been developed for detecting seizure onset by electro-encephalography and triggering in real-time optogenetic silencing of the focus activity for abortion of electrographic and behavioral seizures[27–30]. These external closed-loop systems, however, are invasive and problematic to apply chronically to patients suffering from intractable epilepsy. An alternative idea would be to bring the negative feedback loop to the single neuron level by expressing a sensor for hyperactivity coupled with a molecular actuator able to switch off neuronal activity. An autoregulatory antiepileptic gene therapy, consisting in a glutamate-gated Cl[−] channel that responds to elevations in extracellular glutamate was initially proposed[31]. This biochemical closed-loop molecular device, activated by the excess of endogenous glutamate released during seizures, was effective in a rodent model of focal epilepsy[17,31]. More recently, an alternative transcriptional closed-loop strategy used the promoter of the immediate early gene *c-Fos* as hyperactivity sensor to drive an "*on demand*" transcription of the Kv1.1 channel, previously shown to have potent antiepileptic effects when constitutively expressed by gene therapy[32].

Neuronal hyperactivity and ictal discharges are known to cause significant and rapid changes in intracellular and extracellular pH[33–38]. During seizures, neurons accumulate protons through several mechanisms, including metabolic overproduction of $CO_2/H_2CO_3$ and lactic acid, increased activity of the plasma membrane $Ca^{2+}/H^+$-ATPase, entry of H[+] through ligand/voltage-gated channels, and loss of bicarbonate due to the intense GABA receptor activation[37,39]. The excess of intracellular protons is pumped in the extracellular space predominantly by the $Na^+/H^+$ exchanger (NHE)[40], producing acidic shifts in the local extracellular fluid that have complex effects of neuronal activity. On the one hand, extracellular protons increase excitability by stimulating acid-sensing cationic ion channels (ASICs) in a vicious cycle that tends to maintain hyperexcitability[41–45]. Conversely, they potentiate GABA$_A$ receptor currents, inhibit NMDA receptors and enhance desensitization of AMPA receptors, thus favoring seizure termination[46–48]. The importance of intracellular acidosis in the pathogenesis of paroxysmal activity is underlined by the observation that genetic deletion of $Na^+/H^+$ exchanger causes epilepsy[40]. In a previous work using the ratiometric pH sensor E[2]GFP, whose emission intensity increases in proportion to the drop in pH[49,50] we found significant acidic shifts during epileptic-like hyperactivity that were particularly concentrated at synaptic sites and mostly attributable to the activity of the $Na^+/H^+$ exchanger[51].

Leveraging on the idea that the excess intracellular H[+] generated during hyperactivity can be considered a reliable proxy of upcoming paroxysmal activity at the level of individual neurons[37], in this paper, we have built a synthetic closed-loop chemo-optogenetic nano-machine operating at single neuron level sensitive to intracellular acidosis and fueled by endogenous light. This pH-sensitive luminopsin (pHIL) is composed of an endogenous light generator (RLuc8), the ratiometric pH sensor E[2]GFP and the inhibitory opsin NpHR[52] as a hyperpolarizing actuator to silence neuronal activity. pHIL undergoes Bioluminescence Resonance Energy Transfer (BRET) between the donor RLuc8 and the fluorescent pH sensor E[2]GFP, whose emitted energy is transferred by Fluorescence Resonance Energy Transfer (FRET) to NpHR to activate its chloride-mediated hyperpolarizing current. We found that pHIL is effective in inducing low pH-induced hyperpolarization in HEK293 cells and aborting paroxysmal activity elicited in primary neuronal networks by the administration of convulsants. Moreover, in-vivo expression of pHIL in the hippocampus mitigated acute tonic-clonic seizures induced by pilocarpine and counteracted audiogenic seizures in the PRoline-Rich Transmembrane protein 2 knockout (PRRT2 KO) mouse, a model of genetic epilepsy. The results indicate that pHIL represents an alternative closed-loop gene therapy strategy for drug-refractory epilepsy.

## Results

### E[2]GFP is an excellent sensor of intracellular acidification, a proxy of neuronal hyperactivity

We previously demonstrated that hyperexcitability induced by blockade of inhibitory synaptic transmission with bicuculline (BIC) triggers extracellular acidification at cell bodies and synaptic contacts that is abolished by specific blockade of the $Na^+/H^+$ exchanger[51]. To demonstrate that E[2]GFP can be an efficient sensor of intracellular pH and that neuronal hyperactivity promotes intracellular acidic shifts followed by H[+] extrusion, we engineered a simplified intracellular version of E[2]GFP inserted downstream of the transmembrane sequence of the human cluster of differentiation 4 (CD4) receptor (CD4-E[2]GFP) under the control of the human ubiquitin promoter (Fig. 1A). Confocal microscopy analysis of human embryonic kidney 293 (HEK293) cells and primary hippocampal neurons transfected with the construct revealed the strict membrane localization of CD4-targeted E[2]GFP that in neurons decorates the plasma membrane of the soma and neurites (Fig. 1B, C). Live confocal imaging of transfected primary hippocampal neurons permeabilized to assume various levels of intracellular pH was subsequently used to draw a calibration curve of E[2]GFP emission (405/488 nm excitation ratio; Fig. 1D). Primary hippocampal neurons expressing E[2]GFP on the intracellular side of the membrane and plated on multielectrode arrays (MEAs) were treated with BIC to induce

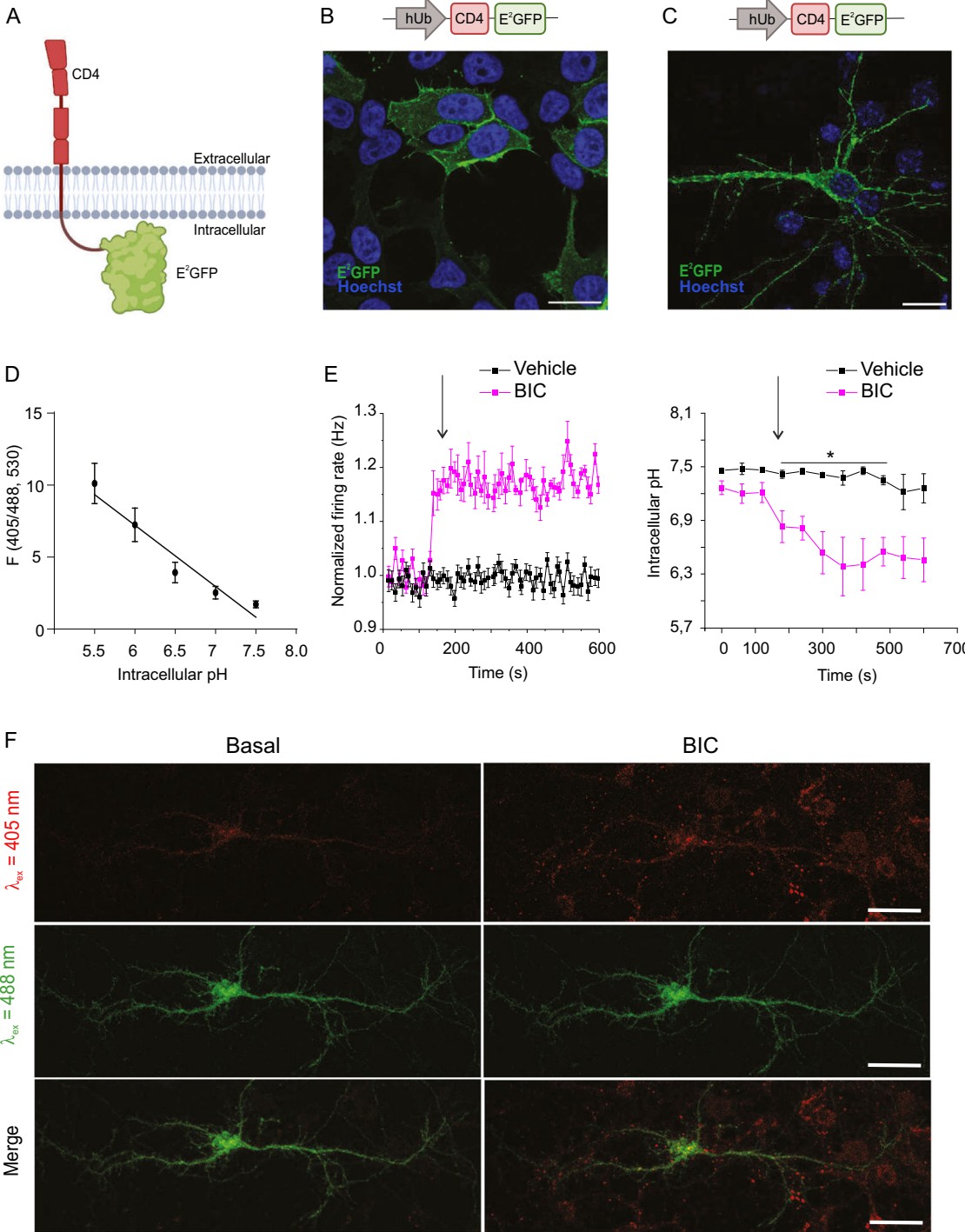

**Fig. 1 | Intracellularly expressed E²GFP sensor reveals the buildup of intracellular acidosis upon sustained hyperactivity in primary hippocampal neurons.**
**A** Schematic illustration of the structure and membrane topology of the CD4-E²GFP intracellular pH sensor (Created with BioRender.com released under a Creative Commons Attribution-NonCommercial-NoDerivs 4.0 International license (https://creativecommons.org/licenses/by-nc-nd/4.0/deed.en)). **B**, **C** Representative confocal images of the membrane localization of CD4-E²GFP in transfected HEK293 cells (**B**) and primary hippocampal neurons (**C**) showing E²GFP fluorescence (green) and Hoechst nuclear staining (blue). In neurons, CD4-E²GFP decorates the membranes of the soma and neurites. Scale bars, 20 μm. **D** Calibration curve of the ratiometric probe obtained in primary hippocampal neurons as a function of intracellular pH. Data are represented by means ± SEM (*n* = 6 neurons from 3 independent preparations). **E** Normalized firing rates of CD4-E²GFP transfected

neurons cultured on MEAs in the presence (red traces) or absence (black traces) of BIC (30 μM; *left*) and the corresponding changes in intracellular pH (*right*). The intracellular pH progressively decreases in discrete spots along cell extensions as neuronal excitability increases. Data are represented by means ± SEM (*n* = 7 and 3 neurons for Vehicle and BIC, respectively from 2 independent preparations, two-sided Mann–Whitney *U*-tests, *$p < 0.05$). **F** Representative fluorescence images of neurons transfected with CD4-E²GFP obtained by E²GFP excitation ($\lambda_{ex}$) at 405 nm (*upper row*, red), 488 nm (*middle row*, green), and the resulting merge images (*lower row*). Images were taken under basal conditions (t = 0 min, *left column*) and upon addition of BIC (30 μM, t = 10 min, *right column*). E²GFP fluorescence increases after BIC treatment upon excitation at 405 nm (upper row), but not upon excitation at 488 nm, demonstrating the proper sensitivity of the pH probe to the pharmacologically induced intracellular acidification. Scale bars, 20 μm.

network hyperactivity and simultaneously recorded for the ratiometric emission of E$^2$GFP and their firing activity. We found that the fast increase in the network firing rate induced by BIC was paralleled by a progressive drop in intracellular pH to ~ 6.3–6.6 (Fig. 1E). The increase in E$^2$GFP emission excited at 405 nm upon network hyperactivity was visible at discrete spots along neurite extensions in time-lapse live confocal microscopy, while E$^2$GFP emission at 488 nm excitation was seemingly constant (Fig. 1F). The results demonstrate that E$^2$GFP is an excellent sensor for the intracellular acidic shifts that accompany paroxysmal electrical activity of the neural network.

## The E$^2$GFP-containing pHIL chimera is targeted to the plasma membrane in HEK293 cells and experiences intramolecular BRET-FRET transitions

Once we demonstrated the performance of E2GFP as an intracellular pH sensor, we aimed to design a homeostatic closed-loop sensor-actuator that could translate the hyperactivity-induced intracellular acidosis into silencing of neuronal activity. Moreover, to avoid the need of external light sources to excite E$^2$GFP and allow a widespread activation of the molecular device, we included RLuc8 as an endogenous light source whose emission can be triggered by the peripheral administration of the substrate coelenterazine (CTZ) and spectrally tuned by RLuc mutations and chemical modification of the substrate[53].

The engineered pH-sensitive luminopsin (pHIL) was a triple chimera composed of: i) RLuc8, acting as an internal light source, ii) the ratiometric pH sensor E$^2$GFP, and iii) NpHR actuating neuronal inhibition (NpHR-E$^2$GFP-RLuc8). In the triple chimera, E$^2$GFP would be excited by BRET from the donor RLuc8 fused to its C-terminus and, under acidic conditions, would act as a donor for the acceptor NpHR fused to the E$^2$GFP N-terminus in a cascade FRET mechanism (Fig. 2A). In addition, an actuator-less E$^2$GFP-RLuc8 chimera was cloned and used as a control (Ctrl). The fusion of RLuc8 and E$^2$GFP provides the proper distance (less than 50 A) to ensure a non-radiative BRET from the RLuc8 donor to the E$^2$GFP acceptor. As the increase in emission of E$^2$GFP triggered by acidic pH occurs when the sensor is excited at ~ 400–410 nm[49], we chose bisdeoxycoelenterazine (CTZ 400a) as RLuc8 substrate to trigger a chemo-emission peaking around 405 nm[53].

HEK293 cells were transfected with either pHIL or Ctrl construct cloned in pcDNA3.0 under the CMV promoter (Fig. 2B). Both constructs were correctly expressed, as evaluated by immunofluorescence for GFP and Western blotting using an anti-luciferase antibody (Fig. 2B, C). Analysis of the subcellular localization of both constructs in live HEK293 cells stained with the membrane marker CellMask revealed the cytosolic expression of the Ctrl chimera, and the co-localization of the pHIL probe with the CellMask-positive plasma membrane, as expected by the presence of the integral membrane protein NpHR (Fig. 2B). Quantification of the expression levels revealed a molecular mass of 62 kDa for Ctrl and 93 kDa for pHIL, with a higher expression of the soluble Ctrl construct in comparison with the NpHR-containing membrane bound pHIL (Fig. 2C).

The activation of pHIL is based on a resonant energy transfer cascade that culminates in the activation of NpHR and the consequent hyperpolarization of the neuron. To test the correct activation of the probe, we transfected HEK293 cells with either Ctrl or pHIL chimeras and exploited the radiative energy transfer cascades ($\chi_{405}$ and $\chi_{510}$ respectively) by performing live imaging at intracellular pH 6.0 and showing both the bioluminescence of RLuc8 upon CTZ 400a administration, followed by the activation of E$^2$GFP emission (Fig. 2D). To have a more quantitative evaluation of the intramolecular BRET/FRET cascades, we analyzed the emission spectra of RLuc8 alone, Ctrl, and pHIL in extracts of HEK293 cells. Upon CTZ 400a administration, the emission spectra of Ctrl and pHIL-treated cells showed the chemo-emission of RLuc8 peaking at 405 nm and the E$^2$GFP fluorescence peak at 510 nm, consistent with BRET between RLuc8 and E$^2$GFP (Fig. 2E).

We next investigated more quantitatively the emission properties of both luciferase and E$^2$GFP to verify the pH-sensitive behavior of the probes and the occurrence of the resonant energy transfer cascade. The emission of RLuc8 alone at 405 nm displayed the previously described pH dependence when fed with increasing concentrations of CTZ 400a (Supplementary Fig. 1A)[54]. A similar decrease in RLuc8 emission at acidic pH was observed in HEK293 cells expressing either Ctrl or pHIL chimera (Supplementary Fig. 1B), with a larger RLuc8 quenching in pHIL-expressing cells likely due to the larger BRET caused by the concomitant FRET between E$^2$GFP and NpHR. As mentioned above, the emission of E$^2$GFP at 510 nm evoked by excitation at 405 nm increases upon acidification while its emission evoked by excitation at 488 nm remains constant[49]. Consistent with this feature, the 510 nm emission from Ctrl-expressing HEK293 cells excited at 405 nm increased when the pH was lowered from 7.4 to 6.0 (Fig. 2F). On the contrary, no pH-dependent increase in the E$^2$GFP fluorescence was observed in pHIL-expressing cells, presumably due to the concomitant quenching of E$^2$GFP by NpHR through FRET. Figure 2G demonstrates the efficacy of the resonant energy transfer cascade upon acidification as a substantial quenching of E$^2$GFP emission in pHIL-expressing HEK293 cells in response to RLuc8 activation upon CTZ 400a administration.

## Intracellular acidic pH shifts cause marked hyperpolarization in pHIL-expressing HEK293 cells

Once assessed the successful engineering of pHIL and the BRET/FRET mechanisms underlying its activity, we asked whether the pH-sensitive probe could regulate the membrane potential upon acidification in HEK293 cells transfected with the construct and treated to assume various levels of intracellular pH. Firstly, we verified whether NpHR within the triple chimera was physiologically responding to direct excitation in the range of the E$^2$GFP emission wavelength (530 nm, 30 mW/mm$^2$). We found that HEK293 cells transfected with pHIL and recorded by whole-cell patch-clamp in the current clamp configuration underwent a ~ 13 mV hyperpolarization upon direct excitation of NpHR, while no effect was present in Ctrl-transfected cells (Fig. 3A). Then, we tested the activation "*on demand*" of NpHR by evaluating the modulation of the membrane potential in HEK293 cells transfected with pHIL upon administration of CTZ 400a (10 μM) at various intracellular pH (6.0, 7.4 and 8.0; Fig. 3B). In pHIL-transfected cells, but not in Ctrl-transfected cells, administration of the RLuc8 substrate elicited a significant hyperpolarization when cells were brought to pH 6.0, which was comparable with that previously triggered by direct light stimulation of NpHR at neutral pH (Fig. 3C). Such response confirmed the proper activation of the BRET/FRET cascade within pHIL in the presence of both acidification and luciferase substrate. Since E$^2$GFP is sensitive to acidic shifts only if excited at ~ 405 nm, we tested the specificity of the effect by employing a distinct substrate (CTZh; 20 μM) known to evoke light emission by RLuc8 at ~ 475 nm[53] (Fig. 3D). Under these conditions, no significant membrane potential modulation was observed regardless of the pH conditions, as expected from the RLuc8 emission wavelength in the range of the E$^2$GFP isosbestic point (475 nm) at which the probe becomes pH insensitive[49] (Fig. 3E).

## pHIL inhibits firing and bursting activities induced by increased intrinsic excitability in primary hippocampal neurons

To transfer the pHIL closed-loop tool to neurons for both in vitro and in vivo studies, the sequences coding Ctrl and pHIL constructs were cloned in an AAV2/1 vector under the control of the Ca$^{2+}$/Calmodulin-dependent kinase IIα (CaMKIIα) promoter to achieve expression in excitatory neurons (Fig. 4A). AAV2/1-transduced primary hippocampal neurons expressed both synthetic constructs at the expected molecular mass of 62 and 93 kDa for Ctrl and pHIL, respectively, as evaluated by Western blotting analysis. Like in HEK293 cells, confocal microscopy confirmed the cytosolic localization of Ctrl construct and

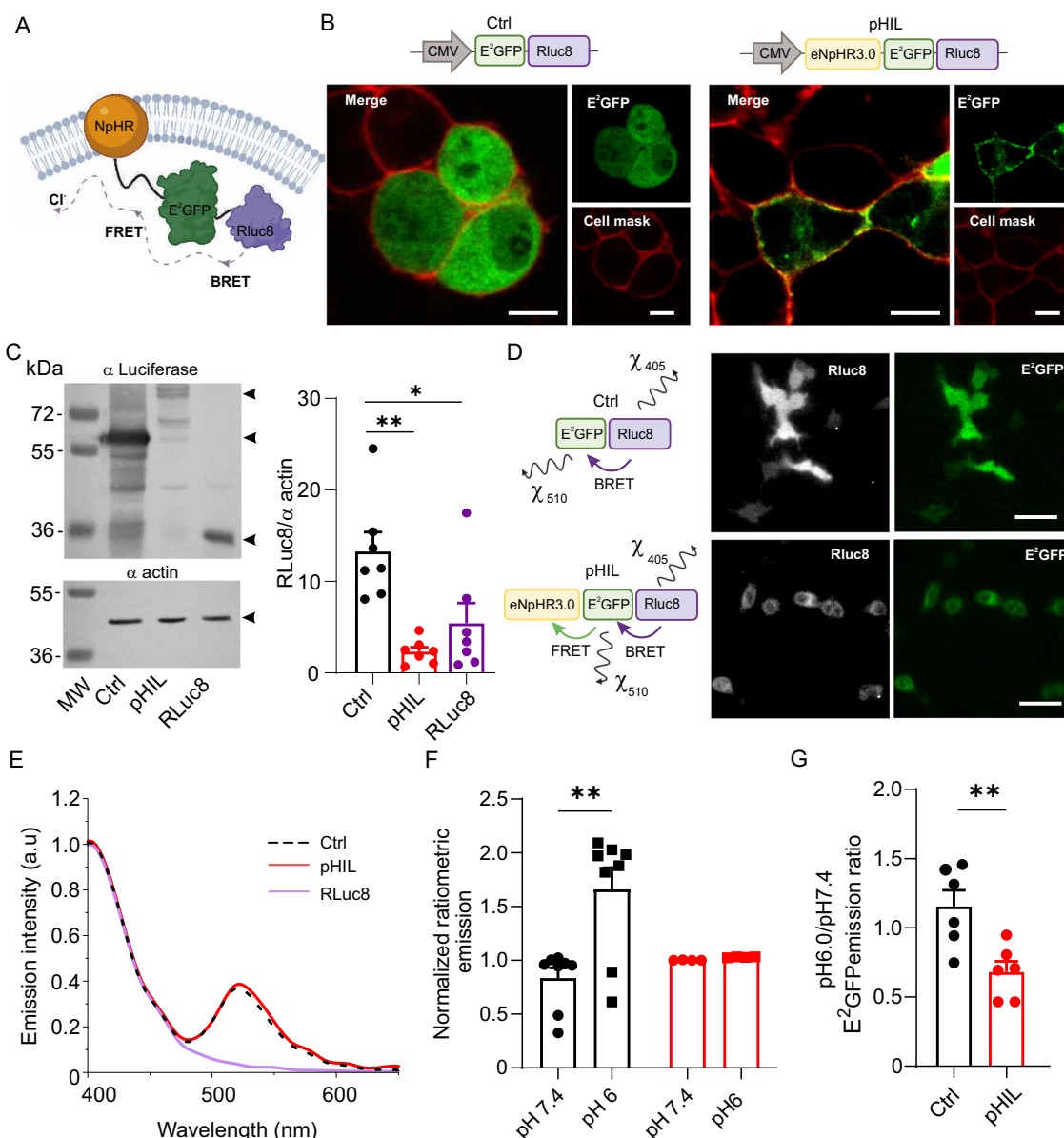

**Fig. 2 | Engineering, expression and BRET/FRET activity of the pHIL chimera.**
**A** Schematic representation of the pHIL structure and cell membrane topology (Created with BioRender.com released under a Creative Commons Attribution-NonCommercial-NoDerivs 4.0 International license (https://creativecommons.org/licenses/by-nc-nd/4.0/deed.en)). **B** Representative live confocal images of membrane staining (red) and the fluorescence of E²GFP (green) showing cytoplasmic localization of Ctrl lacking NpHR (*left*) and the specific membrane localization of pHIL (*right*) in transfected HEK293 cells Scale bars, 20 μm. **C** *Left:* Representative Western blot showing Ctrl, pHIL, and RLuc8 expression in HEK293 cells using an anti-RLuc antibody. Actin immunoreactivity was used as a control of equal loading. Molecular mass markers are reported on the left. The 93, 62, and 34 kDa bands (arrowheads) correspond to pHIL, Ctrl, and RLuc8, respectively. *Right:* The expression of the three constructs in HEK293 cells was assessed as RLuc8/α-actin immunoreactivity ratios ($n = 7$ independent cell preparations, one-way ANOVA $F_{(2,18)} = 9.955$ $p = 0.0012$, Tukey's adjusted p values *$p = 0.0161$, **$p = 0.011$). **D** *Left:* Schematic representation of the resonant and radiative energy transfer of the BRET/FRET mechanisms in Ctrl (*top*) and pHIL (*bottom*) chimeras. *Right:*

Representative live luminescence images of HEK293 cells transfected with either Ctrl (*top*) or pHIL (*bottom*) and brought to pH 6.0 upon administration of CTZ 400a. The images show both RLuc8 bioluminescence and the resulting E²GFP fluorescence activation (2-min integration time). Scale bar, 50 μm. **E** Luminescence emission spectra at pH 6 of HEK293 expressing Ctrl, pHIL, and RLuc8, peaking at 405 nm (luciferase) and 510 nm (E²GFP). Spectra were normalized to the RLuc8 emission peak. **F** The ratiometric emission of E²GFP was calculated at 510 nm upon excitation at 405 and 488 nm and normalized to neutral pH ratio. ($n = 8$ and 4 for Ctrl and pHIL, respectively from 2 cell preparations, one-way ANOVA $F_{(3,20)} = 7.199$ $p = 0.0018$, Tukey's adjusted $p$ value **$p = 0.0014$). **G** The ratio between the E²GFP emissions at pH 6.0 and 7.4 shows a statistically significant decrease in pHIL-transfected HEK293 cells with respect to the controls, demonstrating how the FRET mechanism reduces radiative energy in favor of direct activation of NpHR ($n = 6$ independent cell preparations, unpaired two-sided Student-$t$ test, **$p = 0.0071$). Bar plots depict means ± SEM with superimposed individual experimental points.

the membrane targeting of pHIL (Fig. 4B). The direct excitation of NpHR in pHIL-transduced neurons confirmed the expected hyperpolarizing response with silencing of firing activity (Supplementary Fig. 2). We verified that the expression of either Ctrl or pHIL chimeras in primary neurons did not alter the passive membrane properties,

such as resting membrane potential, membrane capacitance, and input resistance as recorded by whole-cell patch-clamp (Supplementary Fig. 3A–C). Similarly, the intrinsic excitability of pHIL-transduced neurons was virtually unaffected with respect to Ctrl-expressing neurons as shown by the similar rheobase and firing activity as a function

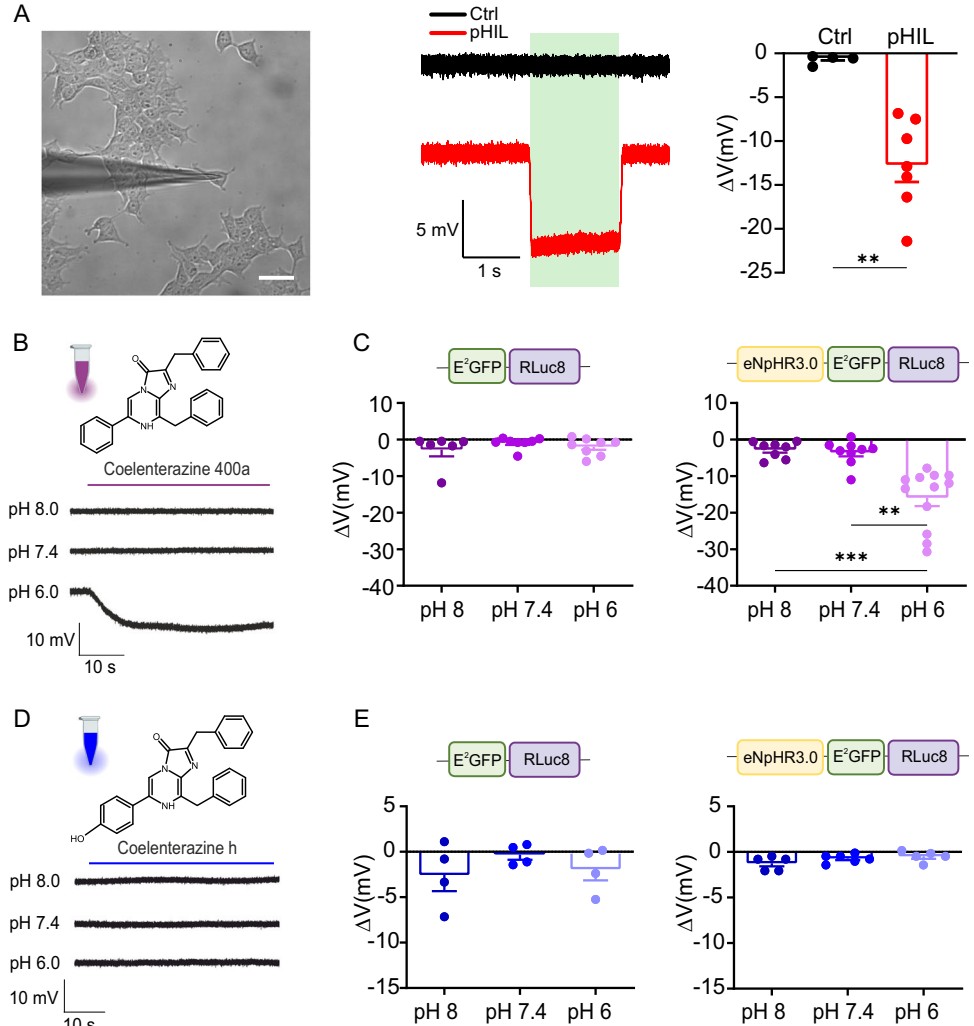

**Fig. 3 | pHIL activation by CTZ 400a hyperpolarizes HEK293 cells. A** *Left and middle:* Representative image (scalebar, 20 μm) and current clamp traces from HEK293 cells expressing Ctrl and pHIL chimeras upon illumination at 530 nm. *Right:* Mean ± SEM hyperpolarization (ΔV) induced by excitation at 530 nm of Ctrl- and pHIL-expressing cells. The significant hyperpolarization of cells expressing pHIL demonstrates the correct activity and photoactivation of NpHR (*n* = 5 and 7 cells for Ctrl and pHIL respectively from 3 preparations, two-sided Mann-Whitney *U*-test, **\**p* = 0.0025). **B, C** Current-clamp experiments on HEK293 cells transfected with pHIL upon administration of CTZ 400a (10 μM) show the marked hyperpolarization at intracellular pH 6.0. (**B**). Comparison of the pH-dependent hyperpolarization (means ± SEM) in HEK293 cells transfected with either Ctrl (**C**, *left*) or pHIL (**C**, *right*). A statistically significant hyperpolarization is apparent only in pHIL-expressing cells

at acidic pH (*n* = 6, 8, 8 cells for Ctrl and *n* = 8, 8, 12 cells for pHIL at pH 8, 7.4 and 6, respectively from 3 preparations, Kruskal-Wallis test *p* < 0.0001, Kruskal-Wallis statistic=18.86, **\**p* = 0.0017, \*\**\**p* < 0.0004). **D, E** Current-clamp experiments on HEK293 cells transfected with pHIL upon administration of CTZh (20 μM) show no response to acidification (**D**). Comparison of the pH-dependent hyperpolarization (means ± SEM) in HEK293 cells transfected with either Ctrl (**E**, *left*) or pHIL (**E**, *right*), the lack of effect when E²GFP is excited at a pH-insensitive wavelength (*n* = 4, 4, 4 cells for Ctrl and *n* = 5, 6, 5 cells for pHIL at pH 8, 7.4 and 6, respectively from 3 independent preparations). Panel B and D (Created with BioRender.com released under a Creative Commons Attribution-NonCommercial-NoDerivs 4.0 International license (https://creativecommons.org/licenses/by-nc-nd/4.0/deed.en)).

of the injected current (Supplementary Fig. 3D–F). As previously reported[11], the expression of NpHR-containing pHIL in neurons did not alter their response to GABA (100 μM), indicating the absence of substantial changes in the chloride equilibrium potential due to background pumping activity (Supplementary Fig. 4).

Once assessed the proper engineering of the viral vector for both Ctrl and pHIL chimeras and ruled out any effect on the electrophysiological properties of the treated cells, we asked whether pHIL could rescue hyperactivity induced by an increase of intrinsic excitability in primary neurons. To this end, we treated Ctrl- or pHIL-transduced primary hippocampal neurons with 4-aminopyridine (4AP), a blocker of voltage-gated K⁺ channels. Neurons were recorded under basal conditions, after addition of 4AP (100 μM), and finally after administration of either CTZ 400a or vehicle to investigate the rescue from hyperactivity (Fig. 4C). Cell-attached configuration was

employed to avoid dialysis of the neuronal cytoplasm during standard whole-cell recordings that would in fact dampen any intracellular acidification and consequently impair the activation of the probe.

The firing rate of single neurons infected with either construct was significantly increased upon 4AP administration. Notably, in pHIL-transduced neurons, the subsequent RLuc8 activation by CTZ 400a was able to return the firing rate to baseline (Fig. 4D). The silencing of the neuronal firing activity induced by 4AP occurred very rapidly, in ~2 s after the addition of CTZ 400a (Supplementary Fig. 5). The comparison between the firing rate of neurons treated with either vehicle or CTZ 400a, normalized to the firing activity induced by 4AP shows indeed a rescue of the hyperactivity in pHIL-transduced neurons (Fig. 4E). In contrast, RLuc8 activation by CTZ 400a was unable to affect the 4AP-induced hyperactivity in neurons transduced with either the cytosolic Ctrl construct (Supplementary Fig. 6A, B) or with a functionally silent

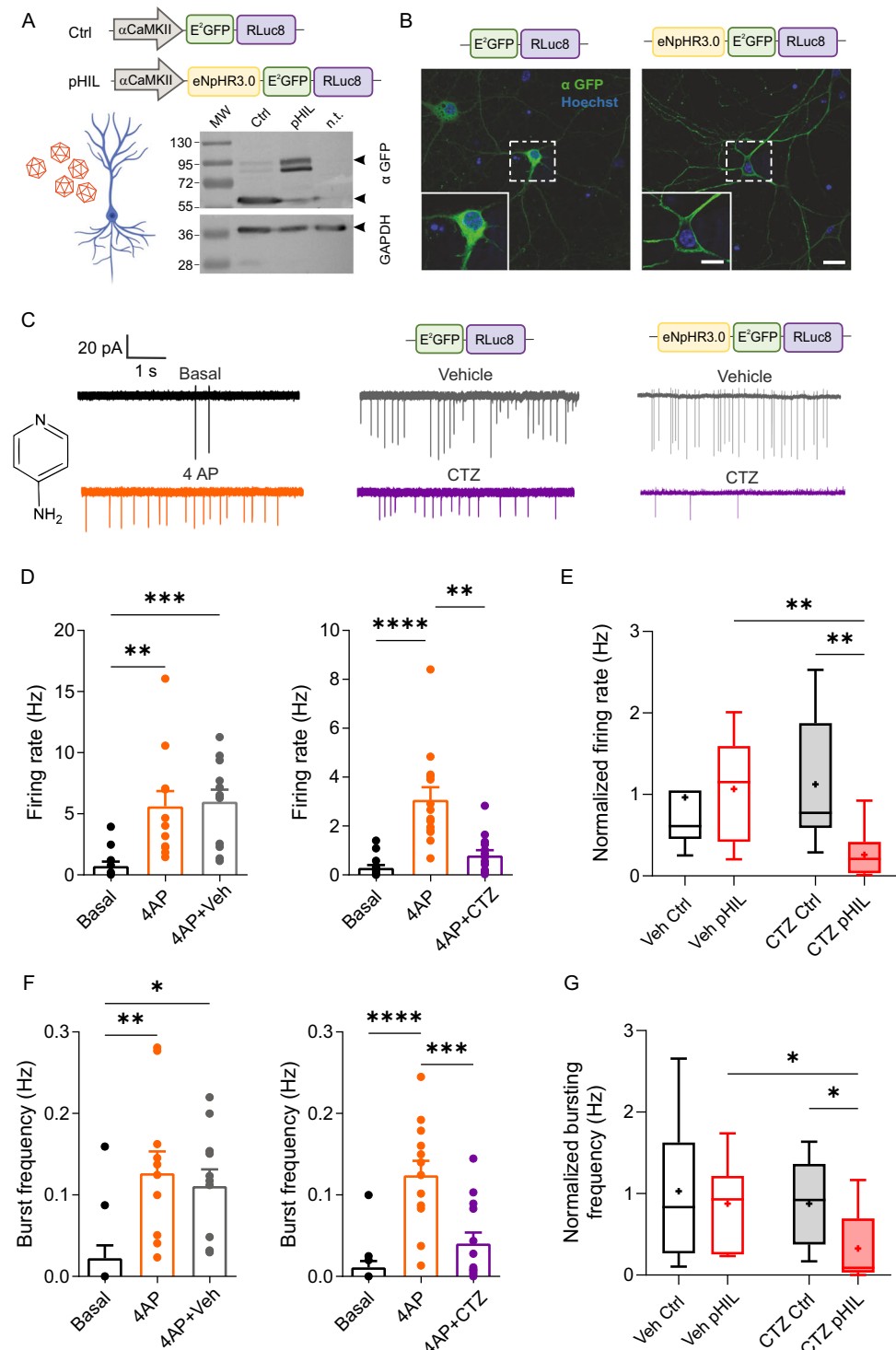

membrane-anchored Ctrl-CD4 construct characterized by the same cellular distribution of pHIL (Supplementary Fig. 6A, B, Supplementary Fig. 7). We then investigated the network activity of the same neurons by evaluating their bursting frequency. Similarly, upon administration of CTZ 400a, the pH-sensitive probe fully rescued the 4AP-induced bursting (Fig. 4F, G), while the expression of either the cytosolic Ctrl construct (Supplementary Fig. 6A, B) or the membrane-anchored Ctrl-CD4 construct (Supplementary Fig. 7) was totally ineffective.

Overall, these results demonstrate the potential of the engineered chemo-optogenetic tool to be switched-on in an activity-dependent manner, when fueled by the pharmacological activation of the bioluminescence.

## pHIL inhibits firing and bursting activities induced by suppression of fast inhibitory transmission in primary hippocampal neurons

We next investigated whether pHIL could recover the epileptic-like paroxysmal firing and bursting activity induced by the blockade of GABA$_A$ receptors with bicuculline (BIC) in primary neurons. Primary hippocampal neurons transduced with either Ctrl or pHIL were recorded by cell-attached voltage-clamp recordings under basal conditions, after BIC (30 μM) administration, and after the successive administration of CTZ 400a or its vehicle (Fig. 5A). The activation of pHIL by CTZ 400a was able to virtually abolish BIC-induced firing and bursting activities, while the hyperactivity of pHIL-expressing neurons

**Fig. 4 | pHIL prevents primary hippocampal neurons from hyperactivity induced by increased intrinsic excitability. A** Schematic illustration of the AAV2/1 vectors encoding pHIL and Ctrl under the CaMKIIα promoter (Created with BioRender.com released under a Creative Commons Attribution-NonCommercial-NoDerivs 4.0 International license (https://creativecommons.org/licenses/by-nc-nd/4.0/deed.en)). Transduced primary hippocampal neurons expressed both fusion proteins, as shown by Western blotting analysis using anti-GFP antibodies and anti-GAPDH antibodies as loading control. **B** Representative confocal images showing the cytoplasmic localization of the Ctrl probe (*left*) and the membrane targeting of pHIL (*right*) in primary neurons (E²GFP, green and Hoechst, blue). *Inset:* a higher magnification of the soma of the neurons highlighted with the dotted square. Scale bar, 20 μm (inset, 10 μm). **C.** Cell-attached voltage-clamp traces from neurons transduced with either Ctrl or pHIL in Tyrode solution (basal) and after the sequential additions of the K⁺-channel blocker 4AP (100 μM) and either CTZ 400a (CTZ; 40 μM) or vehicle (Veh). **D, F** The mean (± SEM) firing rate (**D**) and bursting frequency (**F**) of pHIL-transduced neurons treated with 4AP show the hyperexcitability induced by the convulsant, which is rescued upon infusion of CTZ (firing rate: $n = 12$, Friedman's tests $p = 0.0001$, Friedman statistic=18.17, Dunn's multiple comparisons test **$p = 0.0016$, ***$p = 0.0003$ and $n = 14$, $p < 0.0001$, Friedman statistic=24.17, Dunn's multiple comparisons test **$p = 0.0075$, ****$p < 0.0001$; burst frequency: $n = 11$, one-way ANOVA F (2,30) = 6.804, $p = 0.0037$; Tuckey's adjusted p values *$p = 0.0183$, **$p = 0.005$ and $n = 13$, one-way ANOVA F (2,36) = 18.77, $p < 0.0001$; Tuckey's adjusted p values ***$p = 0.0003$, ****$p < 0.0001$; for 4AP+Veh and 4Ap + CTZ, respectively). **E, G** Box plots showing the changes in firing rate (**E**) and bursting frequency (**G**) in neurons expressing either Ctrl or pHIL upon Veh/CTZ administration, normalized to the activity in the presence of 4AP. The plot shows a reduction of hyperactivity induced by the activation of the pHIL sensor-actuator. Box plots are characterized by median (centre line), mean (cross symbol), 25th and 75th percentiles (box bounds), and min-max values (whisker length) (firing rate: $n = 11, 12, 11, 14$, Kruskal–Wallis test $p = 0.0002$, Kruskal-Wallis statistic=19.38; Dunn's multiple comparisons test **$p = 0.0012$ (CTZ Ctrl vs CTZ pHIL), **$p = 0.0018$ (Veh pHIL vs CTZ pHIL); burst frequency: $n = 10, 11, 11, 13$, Kruskal–Wallis test $p = 0.0064$, Kruskal-Wallis statistic=12.31; Dunn's multiple comparisons test *$p = 0.0311$ (CTZ Ctrl vs CTZ pHIL), **$p = 0.0238$ (Veh pHIL vs CTZ pHIL); for Veh/Ctrl, Veh/pHIL, CTZ/Ctrl and CTZ/pHIL groups, respectively). Unless specified otherwise, n indicates the number of patched neurons from 3 independent preparations.

treated with vehicle was unaffected (Fig. 5B–E). On the contrary, no effect on the paroxysmal activity induced by BIC was observed in Ctrl-transduced neurons, irrespective of the administration of CTZ 400a or its vehicle (Fig. 5C, E; Supplementary Fig. 6C, D).

To analyze the pHIL-induced inhibition of hyperactivity at the network level, we performed extracellular recordings of firing activity in primary hippocampal networks plated on MEA chips and transduced with either pHIL or Ctrl (Fig. 5F, *left*). The mean network firing rates (MFR) recorded under basal conditions, upon addition of BIC (30 μM) and after the subsequent administration of CTZ 400a or vehicle were individually normalized to the baseline MFR, yielding values that can be represented in real time on a pseudo-color scale (Fig. 5F, *right*). In both Ctrl- and pHIL-transduced networks, the blockade of fast synaptic inhibition resulted in a significant 2-fold increase of MFR over baseline that was not altered by the subsequent addition of vehicle. Notably, the addition of CTZ 400a fully normalized the physiological baseline levels of MFR in pHIL-expressing networks, but not in Ctrl-expressing preparations (Fig. 5G). These experiments indicate that pHIL effectively responds to intracellular acidification due to seizure-like activity by inhibiting paroxysmal activity and returning the firing rate to the physiological range, suggesting a potential application as a therapeutic homeostatic tool.

### pHIL is selectively expressed in vivo in principal neurons of the hippocampus

To approach the efficacy of pHIL in ameliorating seizure manifestations in vivo, we firstly assessed the expression of the constructs in excitatory neurons of the hippocampus, the main brain area involved in drug-resistant temporal lobe epilepsy[55]. The AAV2/1 preparation encoding either Ctrl or pHIL under the control of the CaMKIIα promoter was stereotaxically injected in both dorsal hippocampi of 6-month-old wild type mice (Fig. 6A, *left*). One month after the injection, mice were transcardially perfused and analyzed by confocal microscopy exploiting the intrinsic E²GFP emission to evaluate the expression of the probes. The constructs were mostly expressed by the pyramidal cell layer of the dorsal hippocampus with minor spread to the inner cortical layers (Supplementary Fig. 8A). The quantification of the expression of both constructs by Western blotting indeed showed a much higher expression in the hippocampus than in the cortex, while the expression in the cerebellum, used as negative control, was negligible (Supplementary Fig. 8B). To verify that the expression of both pHIL and Ctrl constructs was restricted to excitatory neurons, we imaged E²GFP fluorescence in hippocampal sections stained with either anti-CaMKIIα or anti-parvalbumin (PV) antibodies to identify excitatory and fast-spiking inhibitory neurons. The representative confocal fluorescence images and the quantitative analysis of the colocalization of E²GFP fluorescence with the specific forebrain excitatory neuron marker CaMKII revealed a highly specific expression of both Ctrl and pHIL in excitatory neurons, with overlaps of 71% and 51%, respectively (Supplementary Fig. 8C). A negligible expression of either construct was found in the PV-positive neuronal population, testifying the high specificity of pHIL expression in excitatory principal neurons of the dorsal hippocampus.

### Activation of pHIL in the hippocampus by coelenterazine does not affect cognitive behavior

To examine the effects of pHIL in vivo, we preliminarily evaluated the effects of pHIL activation by time-lapse whole-body bioluminescence and behavioral analysis (Fig. 6A). We intravenously (i.v.) administered 0.3 mg/kg of CTZ 400a to mice transduced with either Ctrl or pHIL one month before the experiment and acquired images every 5 min after the intravenous injection. Representative bioluminescence images of both Rluc8 and E²GFP signals show that the emission was localized in both hemispheres, with a maximum corresponding to the hippocampal formation. Both Ctrl- and pHIL-transduced animals showed a similar extent and timing of luminescence emission, characterized by a peak after the first 5 min after CTZ 400a administration, followed by a progressive reduction in the following 10 min (Fig. 6B).

Next, we performed a CTZ 400a dose-response (0.15, 0.3 and 0.6 mg/kg; i.v.) in pHIL-transduced mice and analyzed it by whole-body bioluminescence emission. Representative images of pHIL-transduced mice captured 5 min after the administration show a progressively increasing emission in parallel with the CTZ 400a dose. The bioluminescence peaked 5 min after the injection and exponentially decayed to become undetectable by whole body imaging 30 min after the administration (Fig. 6C).

Finally, we asked whether the expression of the two probes in the hippocampus could impact on anxiety, locomotor or cognitive performances. To this aim, we compared the behavioral performances of Ctrl- and pHIL-transduced animals with untreated mice in the open field, novel object recognition and contextual fear conditioning tests. Anxiety levels, evaluated as the permanence time in the border versus the center of the open field, and motor performances, evaluated as travelled distance and mean speed, were unaltered (Supplementary Fig. 9A). In the novel object recognition task, Ctrl and pHIL mice displayed comparable levels of object exploration in both the familiarization and the recognition phases, with a similar ability to discriminate between a familiar object and a novel unfamiliar one (Supplementary Fig. 9B). Accordingly, in the contextual fear conditioning test, performed to specifically evaluate the hippocampal

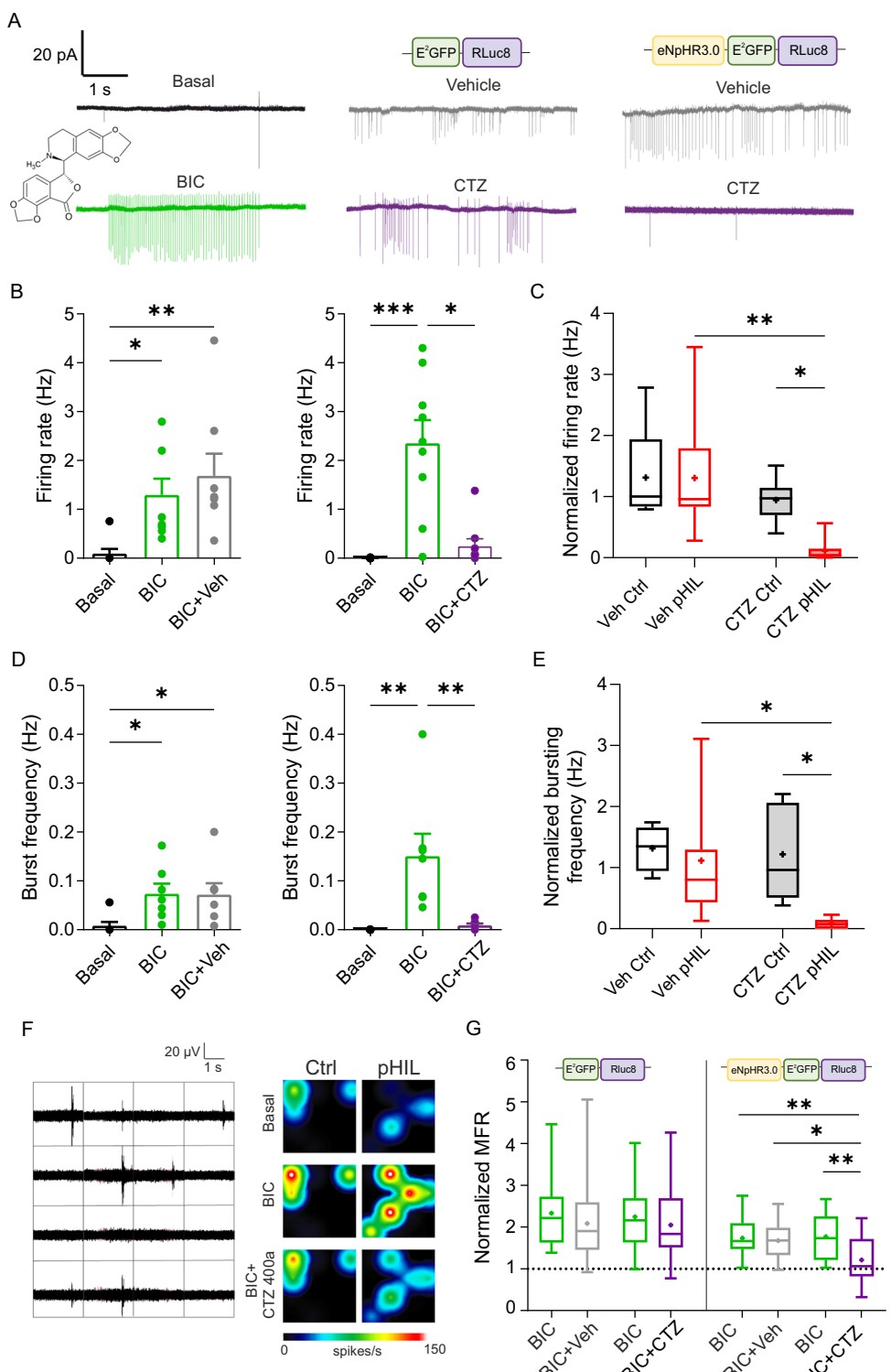

memory functions, Ctrl and pHIL mice showed comparable levels of freezing behavior in all stages of the test, and the ability to recognize a negative environment, such as the cage where they previously received foot shocks Supplementary Fig. 9C).

Although CTZ 400a should activate pHIL and the related NHpR outward current only under conditions of intracellular acidosis, it was important to ascertain that CTZ 400a per se and/or a pHIL background current under physiological pH conditions were not affecting behavioral performances. Similarly to what observed above, the administration of the two highest CTZ 400a doses (0.3 and 0.6 mg/kg) did not

alter motor performances, novel object recognition or contextual fear conditioning (Fig. 6D–F and Supplementary Fig. 10A–C). In the novel object recognition test, the absence of a detectable effect of CTZ 400a was confirmed both when the administration was performed immediately before familiarization and when it was performed before recognition (Fig. 6E and Supplementary Fig. 10B). Similarly, in the contextual fear conditioning test, CTZ 400a was devoid of any behavioral effect both when the administration was performed immediately before the training session and when it was performed before the fear context session (Fig. 6F and Supplementary Fig. 10C).

**Fig. 5 | pHIL prevents primary hippocampal neurons from hyperactivity induced by blockade of fast inhibitory transmission. A** Cell-attached voltage-clamp traces from primary neurons transduced with either Ctrl or pHIL in Tyrode solution (basal) and after the additions of BIC (30 µM) and either CTZ 400a (CTZ; 40 µM) or vehicle (Veh). **B, D** The mean (± SEM) firing rate (**B**) and bursting frequency (**D**) of pHIL-transduced neurons treated with BIC show the paroxysmal hyperactivity after administration of the convulsant, rescued upon CTZ (firing rate: $n = 8$, Friedman's tests $p = 0.0003$, Friedman statistic=13, Dunn's adjusted p values *$p = 0.0373$, **$p = 0.0014$ and $n = 9$, $p < 0.0001$, Friedman statistic=15.94, Dunn's adjusted $p$ values *$p = 0.04$, ***$p = 0.0003$; burst frequency: $n = 7$ one-way repeated measures ANOVA $F_{(2,12)} = 6.434$ $p = 0.0126$, Tukey's adjusted p values *$p = 0.0213$ (Basal vs BIC), *$p = 0.0247$ (Basal vs Veh) and $n = 8$ one-way repeated measures ANOVA $F_{(2,12)} = 9.948$ $p = 0.0028$, Tukey's adjusted p values **$p = 0.0048$ (Basal vs BIC), **$p = 0.0074$ (Basal vs CTZ); for BIC+Veh and BIC + CTZ, respectively). **C, E** Box plots showing the changes in firing rate (**C**) and bursting frequency (**E**) in neurons expressing either Ctrl or pHIL upon Veh or CTZ administration, normalized to the activity in the presence of BIC. The plot shows a reduction of hyperactivity induced by the BRET/FRET cascade mechanism. The plot shows a reduction of hyperactivity induced by the activation of the pHIL sensor-actuator (firing rate: $n = 5, 8, 6, 9$

Kruskal–Wallis test $p = 0.0014$, Kruskal-Wallis statistic = 15.56; Dunn's multiple comparisons test *$p = 0.0256$, **$p = 0.0045$; burst frequency: $n = 4, 7, 5, 8$ Kruskal-Wallis test $p = 0.0031$, Kruskal–Wallis statistic = 13.89; Dunn's multiple comparisons test *$p = 0.0493$ (Veh pHIL vs CTZ pHIL), *$p = 0.0261$ (CTZ Ctrl vs CTZ pHIL); for Veh/Ctrl, Veh/pHIL, CTZ/Ctrl and CTZ/pHIL groups, respectively). **F** Raw firing traces from a representative 16-electrodes MEA well and sample heat maps of the firing activity under basal conditions, in the presence of BIC or BIC + CTZ for primary hippocampal neurons transduced with either Ctrl or pHIL. **G** Box plots showing the changes in mean firing rate (MFR) normalized to the basal condition show a 2-fold increase induced by BIC and the significant decrease occurring in pHIL-expressing neurons upon CTZ administration. The plot shows a reduction of hyperactivity induced by the activation of the pHIL sensor-actuator($n = 31, 23, 27, 21$ electrodes one-way repeated measures ANOVA $F_{(3,84)} = 5.979$ $p = 0.0010$, Tukey's adjusted $p$ values *$p = 0.0124$ (BIC+Veh vs BIC + CTZ), **$p = 0.0022$ (BIC' vs BIC + CTZ), **$p = 0.0038$ (BIC vs BIC + CTZ) for Veh/Ctrl, Veh/pHIL, CTZ/Ctrl and CTZ/pHIL groups, respectively). Box plots are characterized by median (centre line), mean (cross symbol), 25th and 75th percentiles (box bounds), and min-max values (whisker length). Unless specified otherwise, n indicates the number of patched neurons from 3 independent preparations.

Overall, these data show that pHIL expression in the hippocampus and its activation by CTZ 400a do not impact on general hippocampus-dependent behaviors.

### Activation of pHIL by coelenterazine ameliorates pilocarpine-induced tonic-clonic seizures

We finally asked whether pHIL could be efficiently employed to contrast epileptic seizures in vivo. To evaluate whether the homeostatic closed-loop pHIL was able to sense acidosis caused by an incoming seizure and re-establish a physiological excitation/inhibition balance by silencing the activity of hyperexcited principal neurons of the hippocampus, we triggered acute seizures evoked by activation of M1 muscarinic receptors following the systemic administration of the chemo-convulsant pilocarpine.

Mice transduced with either Ctrl- or pHIL one month before the experimental session were subjected to a single intraperitoneal injection of pilocarpine immediately after the administration of CTZ 400a (0.3 mg/kg) to, followed by video-monitoring of the resulting seizures (Fig. 7A). Based on a modified version of the Racine scale, one hour of behavioral observation revealed no differences between Ctrl- and pHIL-treated animals in the first milder stages of the scale, namely immobility, body twitch, and forelimb clonus. However, a significant delay in the appearance of the most severe seizure behaviors, such as Straub tail and generalized tonic-clonic seizures, were observed in pHIL-transduced mice with respect to Ctrl mice (Fig. 7B). In line with these results, the duration of generalized tonic-clonic seizures was significantly reduced by ~ 32 % in pHIL-expressing animals with respect to Ctrl mice (Fig. 7C) and the percentage of mice not developing tonic-clonic seizures was 3-fold higher in the pHIL group than in Ctrl mice (Fig. 7D; Supplementary Movie 1). When the evoked seizures were challenged with increasing doses of CTZ 400a (0.15, 0.3, and 0.6 mg/kg; i.v.), we observed that the anti-seizure effects, quantified in terms of increased latency and decreased duration of tonic-clonic seizures, absent at the lowest dose, were comparable between the two highest doses (Fig. 7E).

Overall, the results demonstrate that the homeostatic activity-dependent pHIL probe has a significant "on demand" anti-seizure activity particularly evident on the most dramatic paroxysmal phenomena that most likely induce pronounced intracellular acidification.

### pHIL counteracts paroxysmal behavior in a genetic model of epilepsy

We finally challenged the anti-seizure activity of pHIL in the PRRT2 KO mouse model of genetic epilepsy[56]. The PRRT2 KO mice does not experience spontaneous seizures but have an abnormally low seizure threshold due to the lack of PRRT2 inhibition on intrinsic and network excitability[57,58] at the level of the hippocampus, where PRRT2 is highly expressed[56]. In this mouse model of human epileptic disorders that include benign familial infantile epilepsy, infantile convulsions with choreoathetosis, and paroxysmal kinesigenic dyskinesia with infantile convulsions, seizures can be easily triggered by environmental stimuli such as sound. Thus, we transduced PRRT2 KO mice with pHIL in the hippocampus and tested the effects of CTZ 400a administration on reflex seizures evoked by sound (Fig. 8A, B). Administration of CTZ 400a significantly increased the latency to paroxysmal manifestations and markedly inhibited or even suppressed the explosive convulsive seizures evoked by sound, by dramatically decreasing the duration of wild running, jumping and tonic-clonic attacks that characterize the PRRT2 KO seizure manifestations (Fig. 8C; Supplementary Movie 2).

The results clearly demonstrate that pHIL is highly effective in suppressing or greatly weakening the epileptic paroxysms in adult PRRT2 KO mice and suggest that it could be a seizure-preventive treatment in chronic epilepsy forms in which seizures are triggered by sensory stimuli.

## Discussion

The excitation/inhibition imbalance that occurs in focal onset seizures puts neurons within the epileptic focus in a metastable state that can generate paroxysmal activity, eventually propagating to the surrounding healthy tissue. As prevention of epileptogenesis has been unsuccessful thus far, the objective of anti-epileptic therapy is to stabilize neurons in the focus and abort the initial focal seizure, or to surgically resect the epileptic area when it does not overlap with eloquent cortical areas.

The ideal intervention would be targeting and stabilizing only epileptogenic neurons, without affecting the activity of healthy neighboring neurons. This has been attempted with gene therapy, but with constitutive and irreversible effects. To overcome these limitations, a closed-loop system detecting the incoming seizure and activating a negative feedback response to silence the source neurons would be the optimal strategy. Gene therapy with optogenetics or chemogenetics with inhibitory DREADDS solves in principle these problems but requires continuous monitoring to administer light or drugs on-demand[17,59]. Indeed, several groups engineered such closed-loop systems in which the early electrographic signs of an incoming seizure detected by scalp electrodes trigger an activation of optogenetically transduced focal neurons[27–30]. These burdensome "externally closed-loop" systems can effectively be replaced by internal closed-loop sensor-actuators working at the level of single neurons or restricted neuronal networks that biochemically detect a proxy of

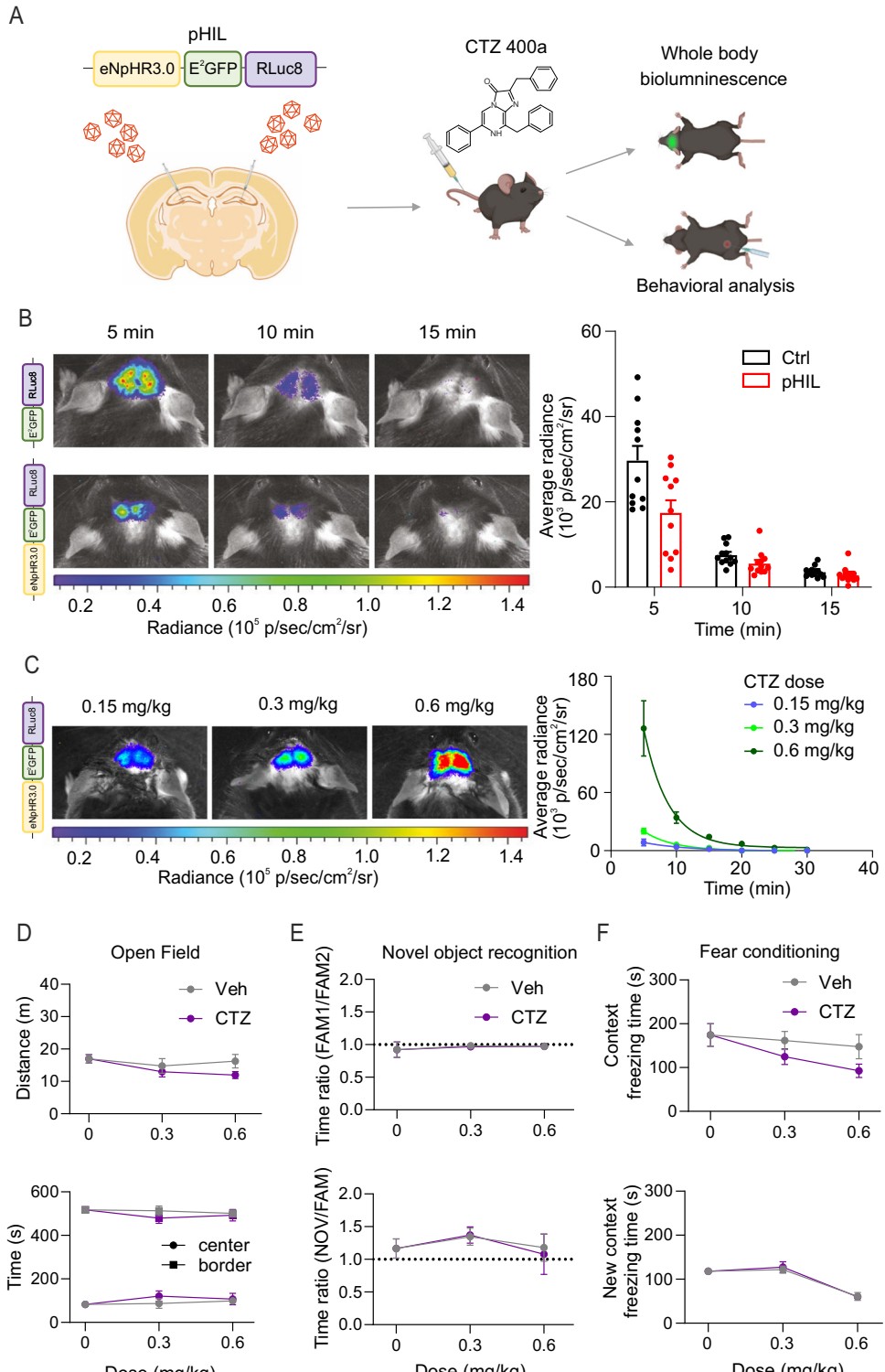

incoming seizures and actuate a homeostatic response to decrease neuronal excitability.

In this work, we have built a chemo-optogenetic, closed-loop pH sensor-actuator chimera fueled by endogenous RLuc8-generated light and using BRET/FRET to activate the inhibitory opsin NpHR[21,51,53,60]. We demonstrated that, upon administration of the luciferase substrate to switch light on at the appropriate wavelength, pHIL is effective in counteracting paroxysmal neuronal activity in vitro and in vivo. pHIL has the advantage of not being a non-specific seizure suppressor, but rather a neuronal homeostatic normalizer that restores the physiological excitation/inhibition balance "*on demand*". Although transduced in all principal neurons, when fueled by CTZ, the closed-loop response of pHIL is only evoked in the hyperactive pathological neurons that experience intracellular acidosis, leaving the healthy bystander cells unaffected. This strategy opens the possibility of a potential translation to a wide range of patients with drug-resistant epilepsy, irrespective of the specific cause (genetic or non-genetic) that had led to the disease. Moreover, the proposed approach could be valuable also under conditions of excessive glutamate release and excitotoxicity that occur acutely in ischemic and traumatic brain

**Fig. 6 | Systemic administration of CTZ 400a in pHIL transduced mice triggers endogenous light emission in the hippocampus without affecting hippocampus-dependent behavior. A** Schematics of the bilateral stereotaxic injection of AAV2/1, encoding either Ctrl or pHIL under the CaMKIIα promoter, in the dorsal hippocampus of 2-month-old wild type C57BL/6 mice followed, one month later, by whole-body bioluminescence imaging and behavioral tests after vehicle or CTZ 400a (CTZ) administration (Created with BioRender.com released under a Creative Commons Attribution-NonCommercial-NoDerivs 4.0 International license (https://creativecommons.org/licenses/by-nc-nd/4.0/deed.en)). **B** *Left:* Representative whole-body bioluminescence images (merged emission of both RLuc8 and E²GFP) of Ctrl (*top row*) and pHIL (*bottom row*) transduced mice acquired every 5 min after the intravenous injection of CTZ (0.3 mg/kg). Radiance intensity is shown in pseudocolors. *Right:* The quantitative evaluation of the mean (± SEM) RLuc8/E²GFP live average radiance values (means ± SEM) in Ctrl (*black bars*) and pHIL (*red bars*) transduced mice displays an early emission peak 5 min after CTZ administration, followed by a progressive decrease ($n = 11$ mice for both Ctrl and pHIL). **C** *Left:* Representative whole-body bioluminescence images acquired 5 min after the administration of increasing doses (0.15, 0.3 and 0.6 mg/kg) of CTZ to pHIL-transduced mice. *Right:* Corresponding RLuc8/E²GFP live average radiance values (means ± SEM) for the three doses as a function of time after CTZ administration. For further details see panel B ($n = 6, 10, 5$ mice for 0.15, 0.3 and 0.6 mg/kg). **D–F** The locomotor activity and hippocampus-dependent behavior were investigated in pHIL-transduced mice upon administration of either CTZ (0.3 and 0.6 mg/kg) or the respective vehicle. The control condition (dose = 0 mg/kg) refers to untreated pHIL-transduced mice. **D** Open field test. The total distance covered by the mice (*top*) and the time spent in the center or along the border (*bottom*) were comparable under all tested conditions, proving no interference of the pharmaceutical treatment with locomotor activity (means ± SEM of $n = 7, 7, 10$ for 0, 0.3 and 0.6 mg/kg respectively in both Veh and CTZ groups). **E** Novel object recognition. No effects of CTZ were observed both in the familiarization phase (*top*) and in the recognition phase (*bottom*) when mice were exposed to one object previously explored and a novel unfamiliar object (means ± SEM of $n = 7, 7, 7$ for 0, 0.3 and 0.6 mg/kg respectively in both Veh and CTZ groups). **F** Contextual fear conditioning. No significant differences in freezing times were observed both during the fear conditioning phase (*top*) and in control exposure to a new context (*bottom*) (means ± SEM of $n = 7, 7, 7$ for 0, 0.3, and 0.6 mg/kg respectively in both Veh and CTZ groups). In E and F, either CTZ or vehicle was administered before the novel recognition phase and the conditioning session, respectively. $p > 0.05$, two-way repeated measures ANOVA (D-F).

injuries[61,62] or chronically in neurodegenerative diseases according to the "dying forward' excitotoxicity hypothesis[63,64].

The reliability of intracellular pH shifts as a proxy of hyperactivity at the single neuron level is demonstrated by the progressive intracellular, and successively extracellular, acidification of neurons in hyperactive primary networks subjected to treatment with convulsants or to high-frequency stimulation (this paper,[51,65]) and, under the same conditions, by the efficacy of pHIL in counteracting the paroxysmal firing activity. The role of intra/extracellular pH changes during epileptogenic activity is multifaceted. Protons are precociously generated inside the hyperactive neuron and transferred to the extracellular space mainly by NHE[40,51]. Although extracellular acidosis has been reported to potentiate GABAergic transmission and depress glutamatergic transmission, with a possible role in seizure termination[36,38,42,46–48] NHE KO mice display increased excitability and epilepsy indicating that intracellular acidification, followed by H⁺ extrusion by NHE increases excitability likely through stimulation of ASICs[40].

A few caveats should be considered regarding the pHIL strategy here described. Being the actuator a Cl⁻ pump, the presence of a background current may increase neuronal excitability by altering the Cl⁻ equilibrium potential and the extent of GABA-evoked inhibitory currents, as previously described[66]. However, in neurons expressing pHIL, GABA transmission is not affected, testifying the absence of a significant Cl⁻ background current in the absence of RLuc activation. Another potential problem is represented by the E²GFP sensitivity to the intracellular concentration of Cl⁻. It is known that the emission of E²GFP, as a member of the class of yellow fluorescent proteins, is quenched by the concentration of anions, namely Cl⁻, in a dose-dependent fashion. By virtue of this property, E²GFP has been used as a dual intracellular pH and Cl⁻ sensor[38,67–69]. In our chimera, we used the short wavelength band of E²GFP excited by the enhanced *Renilla* luciferase RLuc8 (ΔN3-RLuc8-S257G[22]) fueled by CTZ 400a to emit in the 400-410 nm range[53]. At this wavelength, E²GFP emission strongly increases as pH decreases, and is capable to activate NpHR irrespective of potential changes in intracellular Cl⁻. Should the increase in [Cl⁻]$_i$ resulting from prolonged pHIL activation quench E²GFP emission, this would result in a homeostatic rundown of pHIL activity preventing GABA to become excitatory.

Coelenterazine-powered pHIL proved to be effective in vivo in ameliorating seizure activity evoked by the acute administration of the convulsant pilocarpine, as well as seizure manifestations in the PRRT2 KO mouse, a model of human genetic epilepsy. In the pilocarpine-evoked seizures, the effect was predominantly exerted on the expression of the most severe seizure phenotypes, that were delayed in appearance and shortened, as well as on the susceptibility to tonic-clonic seizures that was markedly decreased. Similarly, in the PRRT2 KO mouse pHIL delayed and shortened the sound-triggered paroxysmal attacks and virtually abolished the convulsive events. The specificity of the sensor-actuator probe in attenuating the most severe seizure manifestations is conceivably attributable to the more pronounced pH shifts that lead to a stronger activation of NpHR and silencing of excitatory neuron activity. Although active in counteracting hippocampus-triggered paroxysms, pHIL does not affect hippocampal-dependent behaviors, both under basal conditions and after the administration of CTZ 400a, an important requisite in view of potential future clinical applications.

It is interesting to compare the pHIL strategy described here with two previously reported closed loop "hyperactivity sensor" strategies that are conceptually similar to pHIL, namely: (i) a self-regulating chemogenetic strategy with eGluCl, an optimized *C. elegans* glutamate-gated chloride channel that opens and inhibits neurons upon increases of extrasynaptic glutamate, as it occurs during seizures[31]; and (ii) a transcriptional closed-loop in which a transgenic hyperpolarizing Kv1.1 channel is expressed under the control of the promoter of the immediate early gene c-Fos that is strongly activated by neuronal hyperactivity[32,70]. In particular, the cFos transcriptional strategy needs time to be effective after the initial trigger, being ineffective to abort the first seizure, but protecting against subsequent seizures within the temporal window in which the channel is expressed (∼2 days). On the other hand, the eGluCl strategy is conceptually similar to pHIL. Both strategies are activated by a proxy of neuronal hyperactivity (extrasynaptic spillover of glutamate in the former, intracellular acidosis in the latter) and have the potential drawbacks of expressing a heterologous protein on the neuronal membrane (eGluCl or NHpR), possibly triggering immune responses. Moreover, both strategies used the outward chloride current to actuate inhibition of neuronal activity, which might imply long-term changes in the Cl⁻ reversal potential, making the therapy progressively less effective[66]. Although eGluCl can still work by shunting inhibition in the presence of a decreased electrochemical gradient for chloride, pHIL has a negligible background NHpR activity and its "on demand" Cl⁻-pumping activity does not markedly depend on the chloride gradient. The advantage of eGluCl over pHIL is that it uses an endogenous proxy of hyperactivity and does not require the administration of a drug, although the dynamics of glutamate signaling and spillover can make eGluCl activation not fully predictable. In this respect, pHIL functions only in the presence of CTZ 400a that transiently activates the sensor-

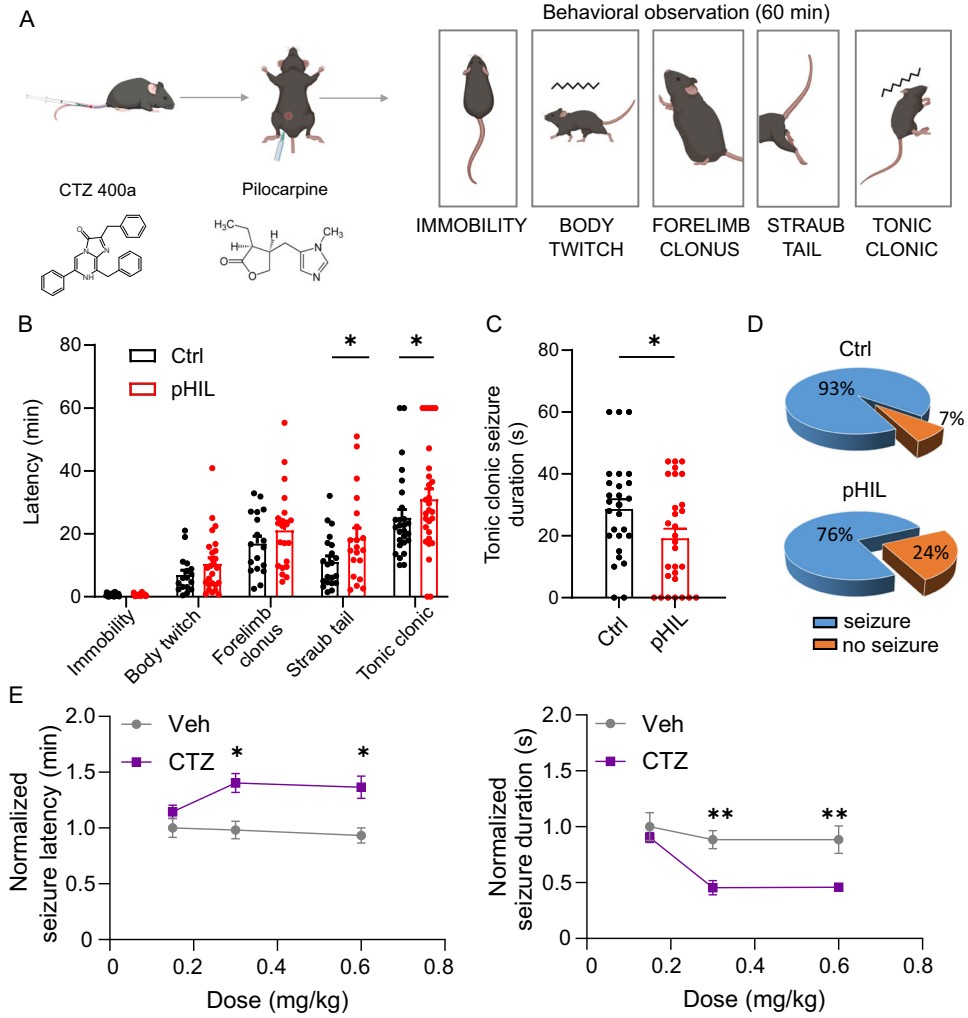

**Fig. 7 | pHIL activation in-vivo by CTZ 400a counteracts pilocarpine-induced seizures. A** Experimental timeline of in vivo experiments to evaluate seizure susceptibility. Both Ctrl- and pHIL-transduced mice were subjected first to CTZ 400a (CTZ) injection to activate the constructs and, immediately after, to the intraperitoneal administration of pilocarpine (300 mg/kg) to induce seizure activity. Mice were videorecorded for 1 h and the seizure phenotype classified according to a modified Racine scale (*right*) (Created with BioRender.com released under a Creative Commons Attribution-NonCommercial-NoDerivs 4.0 International license (https://creativecommons.org/licenses/by-nc-nd/4.0/deed.en)). **B** The latency of most ictal behaviors shows a trend toward mitigation of the epileptic phenotype in pHIL-treated animals, becoming statistically significant for the most severe manifestations including tail dorsiflexion (Straub tail; two-sided unpaired Student's *t*-test *$p = 0.0418$) and generalized tonic-clonic seizures (two-sided unpaired Student's *t*-test *$p = 0.0159$). Bar plots depict means ± SEM with superimposed individual experimental points (Ctrl: $n = 27, 17, 18, 21, 27$; pHIL: $n = 28, 26, 21, 20, 29$ for immobility, body twitch, forelimb clonus, Straub tail and tonic-clonic behaviors, respectively). **C** The mean (± SEM) duration of the tonic-clonic seizures is

significantly decreased in pHIL-expressing mice treated with CTZ ($n = 27, 29$ for Ctrl and pHIL, respectively, two-sided unpaired Student's *t*-test *$p = 0.03$). **D** The analysis of the occurrence of tonic-clonic seizures (expressed as percentage of mice reaching the last stage of the Racine scale) shows an over 3-fold increase in the percentage of seizure-free animals in the pHIL-expressing group with respect to the Ctrl-expressing mice. **E** Dependence of the pHIL anti-seizure effects on the CTZ dose. Increasing doses (0.15, 0.3, and 0.6 mg/kg) of CTZ or the corresponding vehicle volume were administered to pHIL-transduced mice before pilocarpine. The latency (*left*) and duration (*right*) of tonic-clonic seizures normalized to the performance of the lowest vehicle volume are shown as a function of the dose (means ± SEM of $n = 5, 5, 4$ and $5, 5, 5$ for Ctrl and pHIL at 0.15, 0.3 and 0.6 mg/kg, respectively; two-way repeated measures ANOVA; latency: treatment effect $F(1,23) = 24.48$ $p < 0.0001$, Šídák's multiple comparisons tests *$p = 0.0209$ and *$p = 0.0295$ for 0.3 and 0.6 mg/kg, respectively; duration: treatment effect $F(1,22) = 21.19$ $p = 0.0001$; concentration effect $F(2.22) = 7.690$ $p = 0.0029$, Šídák's multiple comparisons tests **$p = 0.0055$ and **$p = 0.0062$ for 0.3 and 0.6 mg/kg, respectively). (Created with BioRender.com).

actuator. Although whole-body imaging showed a time window for light generation of about 20–30 min after CTZ 400a intravenous administration, it is likely that the fully effective BRET/FRET cascade occurs at the single cell level for much longer times having undetectable levels of radiative emission due to a poor tissue penetration of 405 and 510 nm light.

We have developed a gene therapy strategy generating a cell-autonomous homeostatic sensor-actuator that silences neural activity when hyperactivity causes intracellular acidic shifts. It is a chemo-optogenetic pH sensor-actuator powered by endogenous light generation obtained by the systemic administration of a suitable luciferase

substrate. In primary neuronal networks, acute drug-evoked generalized seizures and sound-evoked seizures in a genetic model of epilepsy, pHIL proved to be effective in counteracting the paroxysmal phenotypes. Based on the proof-of-concept demonstrated in this paper, pHIL represents a potentially promising approach to treat drug-refractory chronic epilepsy independent of the specific etiology, particularly when the seizure focus is close to eloquent cortices, which makes the surgical approach unfeasible. To translate pHIL into a seizure-preventive therapy in chronic epilepsy, in which spontaneous seizures occur unpredictably, it will be important to optimize the administration route and pharmacology of CTZ 400a.

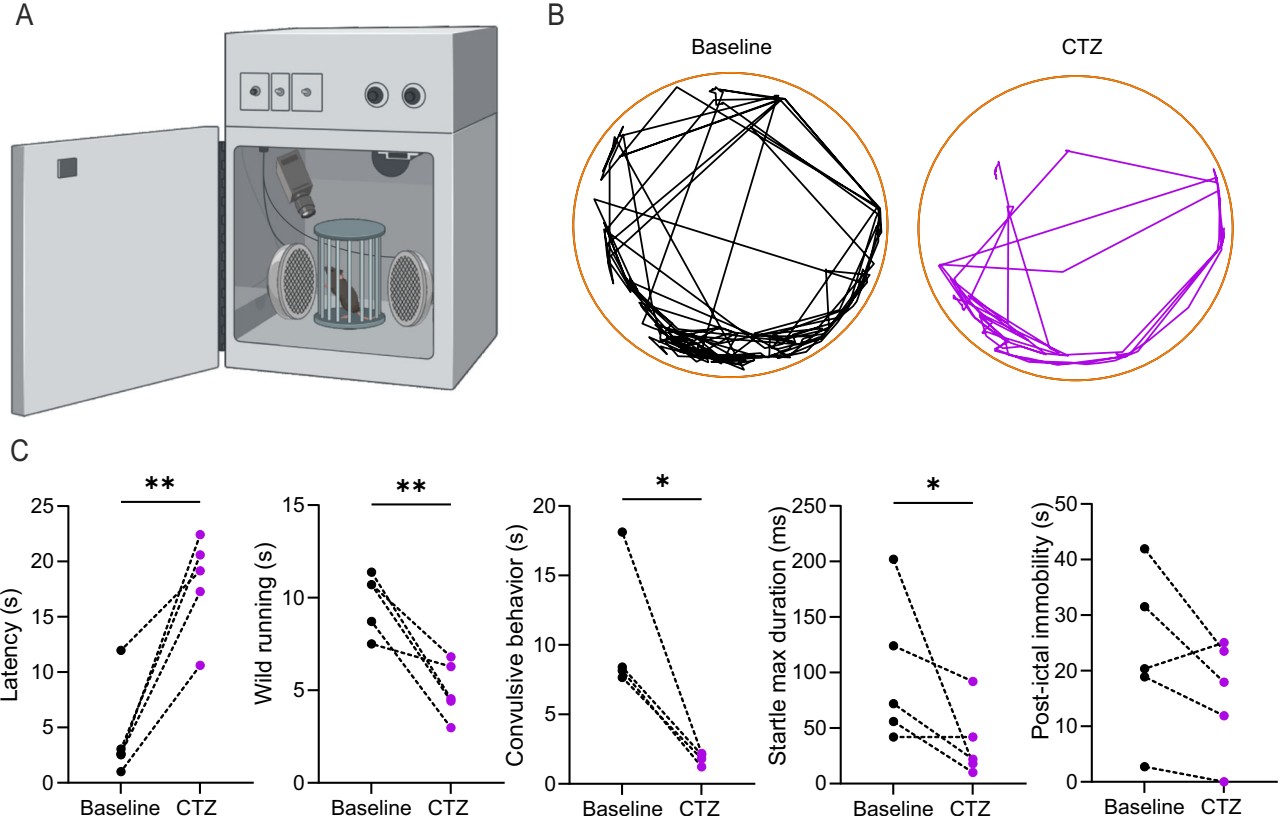

**Fig. 8 | pHIL activation by CTZ 400a counteracts audiogenic seizures in PRRT2 KO mice. A, B** Schematics of the startle setup for triggering and monitoring audiogenic seizures in pHIL-transduced PRRT2 KO mice (**A**) and representative video-tracking of the sound-evoked paroxysmal behavior of the same PRRT2 KO mouse before and after CTZ 400a (CTZ) administration (Created with BioRender.com released under a Creative Commons Attribution-NonCommercial-NoDerivs 4.0 International license (https://creativecommons.org/licenses/by-nc-nd/4.0/deed.en)) (**B**). **C** PRRT2 audiogenic seizures range from wild running to convulsive attacks and jumping, followed by post-ictal immobility. The behavioral analysis, performed before and after CTZ administration (0.3 mg/kg; $n = 5$ mice) shows, from left to right, the latency to paroxysmal attacks (two-sided paired Student's $t$-test **$p = 0.0044$), and the duration of wild running (two-sided paired Student's $t$-test **$p = 0.0092$), convulsive attacks (two-sided paired Student's $t$-test *$p = 0.0356$), jumps and post-ictal immobility (two-sided paired Student's $t$-test *$p = 0.0326$). (Created with BioRender.com).

## Methods

### Experimental animals

All experiments were carried out in accordance with the guidelines established by the European Community Council (Directive 2010/63/EU of 22 September 2010) and were approved by the Italian Ministry of Health (Authorization #73-2014-PR on Dec 5, 2014, 597/2021-PR and 539/2022-PR). Wild type C57BL/6 mice were purchased from Charles River (Calco, Italy). Constitutive PRRT2 KO mice, generated by the IMPC European Consortium at the Sanger Institute (UK) in the frame of the European EMMA/Infrafrontier, were backcrossed to a full C57BL/6 background[56]. To reduce the biological variability only male mice between 2-6 months of age were used in this study. Mice were maintained on a 12:12 h light/dark cycle (lights on at 7 a.m.) at constant temperature ($22 \pm 1\,^{\circ}C$) and relative humidity ($60 \pm 10\%$), provided drinking water and a complete pellet diet (Mucedola, Settimo Milanese, Italy) *ad libitum*, and housed under conditions of environmental enrichment in the IRCCS Ospedale Policlinico San Martino Animal Facility. All efforts were made to minimize suffering and reduce the number of animals used.

### Plasmid construction and virus production

All restriction enzymes were purchased from Promega (Madison, WI) unless otherwise indicated; all plasmids were verified by digestion with restriction enzymes and direct sequencing. Constructs were initially cloned in pcDNA3.1 under the control of the cytomegalovirus (CMV) promoter. The halorhodopsin version eNpHR3.0 and the $\Delta$N3-RLuc8-

S257G sequences were amplified from the pLenti-ILMO2 and pAAV2-ILMO2 constructs[23,26]. The truncated form of E²GFP ($\Delta$N12-E²GFP-$\Delta$C15) was amplified from pcDNA-E²GFP (kindly provided by Dr. R. Bizzarri, Institute of Biophysics, Italian National Research Council, Pisa, Italy).

*CD4-E²GFP.* The truncated form of the E²GFP ($\Delta$N12-E²GFP-$\Delta$C15) was cloned in the hCD4-mOrange plasmid (cat. #110192, Addgene, Watertown, MA) downstream of the CD4 sequence. First, an ER export sequence was inserted downstream of truncated E²GFP (Forward: 5′-atccaagcttatggtgagcaagggcgaggagctg-3′, Reverse: 5′-atccgtagcagaacc cggcggcggtcac-3′). For the second PCR amplification, the $\Delta$N12-E²GFP-$\Delta$C15/ER was used as template (Forward: 5′-atccaagcttatggtgag-caagggcgaggagctg-3′, Reverse: 5′-atccgaattcttacacctcgttctcgtagcagaa-3′). The amplified sequence and the hCD4-mOrange plasmid were digested with the HINDIII/ECORI restriction enzymes and then ligated. The CD4-$\Delta$N12 E²GFP-$\Delta$C15/ER sequence was subsequently cloned in the lentiviral backbone pLenti-Ub-iLMO2, downstream of the promoter of human ubiquitin (hUb). CD4-$\Delta$N12-E²GFP-$\Delta$C15/ER was PCR amplified (Forward: 5′-atccggatccatgaaccgggggagtccc-3′, Reverse: 5′-atccgaattcttacacctcgttctcgtagcagaa-3′). Both pLenti-Ub-iLMO2 and the amplified sequence were digested with BAMHI/ECORI and ligated.

*pHIL and Ctrl constructs.* For the Ctrl construct, the $\Delta$N3-RLuc8-S257G sequence was PCR amplified (Forward: 5′-atccggtaccaaggtg-tacgaccccgag-3′, Reverse: 5′-atccgaattcttacacctcgttctcgtagc-3′). Both pcDNA3.1 and $\Delta$N3-RLuc8-S257G were subsequently digested with ECORI/KPNI and ligated using T4 ligase. The $\Delta$N12-E²GFP-$\Delta$C15 sequence was PCR amplified (Forward: 5′ atccaagcttagggatccatg

gtgagcaagggcgaggagctg-3', Reverse: 5'- atccggtaccccccggcggcggtcac gaac-3'). Both ΔN12-E²GFP-ΔC15 and pcDNA-ΔN3-RLuc8-S257G were digested with HINDIII/KPNI and then ligated to insert ΔN12-E²GFP-ΔC15 upstream of ΔN3-RLuc8-S257G. For the pHIL construct, eNpHR3.0 was PCR amplified (Forward: 5'-atccggatccatgacagagaccctgcctcccG-3', Reverse: 5'-atccgcggccgcatcatcagccggggtc-3'). Both pcDNA3.1 and eNpHR3.0 were subsequently digested with BAMHI/NOTI and ligated using T4 ligase. The ΔN12-E²GFP-ΔC15 – ΔN3-RLuc8-S257G sequence was PCR amplified from the Ctrl plasmid (Forward: 5'- atccgcggccg-cagtgagcaagggcgaggagctg-3', Reverse: 5'-atccctcgaggaattcttacacctcgtt ctcgtagc-3'). Both pcDNA-eNpHR3.0 and the ΔN12-E²GFP-ΔC15 – ΔN3-RLuc8-S257G sequence were digested with NOTI/XHOI and ligated, to insert the ΔN12-E²GFP-ΔC15 – ΔN3-RLuc8-S257G sequence down-stream of eNpHR3.0. Both constructs also included, at the 3' end of the fusion protein, an ER-export sequence to facilitate the ER export of proteins and their insertion in plasma membrane[52]. pHIL and Ctrl cassettes were subsequently cloned into the pAAV2 backbone (from pAAV2-ILMO2) for producing pAAV2-pHIL and pAAV2-Ctrl. To achieve this, the pAAV2 backbone, pcDNA-pHIL and pcDNA-Ctrl plasmids were digested with BAMHI/ECORI and ligated. To reduce the transgene length, the CaMKIIα (1.3 Kb) promoter was replaced by the CaMKIIα mini-promoter (0.3 Kb)[71]. For cloning pAAV2-pHIL, the CaMKIIα mini (0.3 Kb) promoter was amplified from the pAAV2-ILMO2 plasmid (Forward: 5'-atccacgcgtacttgtggactaagtttgtt-3', Reverse: 5'-atccg-gatccgctgcccccagaactagggg-3'). Both CaMKIIα mini (0.3 Kb) promoter and the recipient vector (pAAV2-pHIL) were digested with MLUI/ BAMHI and then ligated. For cloning pAAV2-Ctrl, the CaMKIIα mini-promoter and the ΔN12-E²GFP-ΔC15 – ΔN3-RLuc8-S257G sequence were PCR amplified from pAAV2-pHIL (Forward: 5'-atccacgcgtacttgtg-gactaagtttgtt-3', Reverse: 5'-atccgaattcttacacctcgttctcgtagc-3'). Both CaMKIIα mini (0.3 Kb) - ΔN12-E²GFP-ΔC15 – ΔN3-RLuc8-S257G sequence and the recipient vector (pAAV2-Ctrl) were digested with MLUI/ECORI and then ligated. For cloning AAV2/1 CaMKIIα-Ctrl-CD4, the CD4 sequence was PCR amplified from pLenti CD4-ΔN12-E2GFP-ΔC15/ER plasmid (Forward: 5'-tgctctagacatgaaccggggagtccctt-3', Reverse: 5'-cgcggatccaatggggctacatgtctt-3'). Both CD4 sequence and the recipient vector (AAV2/1 CaMKIIα-Ctrl) were digested with XBAI/ BAMHI and then ligated. AAV2/1 particles (AAV2/1 CaMKIIα-pHIL, AAV2/1 CaMKIIα-Ctrl and AAV2/1 CaMKIIα-Ctrl-CD4) were generated as previously described[72]. Briefly, human embryonic kidney HEK293 cells were co-transfected with the required AAV vector together with the plasmids pRV1, pH21 and pHelper using a $Ca^{2+}$ phosphate method. Forty-eight hours post-transfection, cells were harvested and lysed, and viruses purified over heparin columns (GE HealthCare, Milano, Italy). Viral vectors were titrated at concentrations ranging from $1 \times 10^{11}$ to $1 \times 10^{12}$ transducing units (TU)/ml and used at a multiplicity of infection (MOI) of 10,000. The efficiency of infection, estimated by counting neurons expressing GFP protein with respect to the total number of cells stained with DAPI, ranged between 70% and 90%.

## Cell cultures, transfection and transduction

All cell culture reagents were from Invitrogen (Carlsbad, California), unless otherwise specified. Human embryonic kidney 293 T (HEK293) cells were cultured in Dulbecco's Modified Eagle Medium (DMEM) (cat. #11995-073, Invitrogen) supplemented with 10% fetal bovine serum (FBS), 1% glutamine and 1% penicillin/streptomycin (complete DMEM), in a humidified 5% $CO_2$ atmosphere at 37 °C. For splitting, cells were carefully washed with phosphate-buffered saline (PBS) and incubated with trypsin/EDTA for 1–2 min. Cells were subsequently centrifuged at 3000 x $g$ for 5 min, and then resuspended in fresh medium. HEK293 cells were transfected with using Lipofectamine 2000 following man-ufacturer's recommended protocols.

Hippocampal neurons were obtained from embryonic day 18.5 C57BL/6 mouse embryos. Hippocampi were dissected in ice-cold PBS, incubated with trypsin 0.125% for 15 min at 37 °C, and mechanically

dissociated. Neurons were then resuspended and plated on poly-L-lysine-coated glass coverslips or plastic wells in Neurobasal medium containing 10% fetal bovine serum (FBS), 1% Glutamax and 1% peni-cillin/streptomycin. After 3 h, the medium was removed and replaced with Neurobasal containing 2% B27 supplement, 1% Glutamax and 1% penicillin/streptomycin. Primary hippocampal neurons were either transfected with either lentiviral constructs encoding CD4-E²GFP at 10 DIV and analyzed at 12 DIV or transduced with AAV2/1 particles car-rying pHIL/Ctrl/Ctrl-CD4 constructs at DIV 3 and analyzed at DIV 14.

## Morphological studies

**Immunocytochemistry.** HEK293 cells and hippocampal neurons were plated on poly-L-lysine-coated 18-mm coverslips. Forty-eight hours after HEK293 cell transfection, cells were incubated with the mem-brane marker CellMask deep red (cat. #C10046, ThermoFisher Scien-tific, Waltham, MA). HEK293 cells were incubated with the probe for 5 min, washed with PBS and analyzed for plasma membrane staining with live imaging using a Leica SP8 confocal microscope (Leica Microsystems, Wetzlar, Germany). Ten days after infection, samples were fixed with 4% paraformaldehyde in PBS for 20 min and subse-quently permeabilized for 5 min with 0.2% Triton X-100 in PBS before incubation in blocking buffer (PBS/0.05% Tween-20/3% bovine serum albumin) for 30 min. Cells were then incubated at room temperature (RT) for 1 h with the following primary antibodies: rabbit polyclonal anti-GFP (1:200; cat. #A111222 ThermoFisher Scientific), guinea pig polyclonal anti-GFP (1:200; cat. #132005 Synaptic System GmbH, Göttingen, Germany), rabbit monoclonal anti-Renilla luciferase (1:200; cat. #ab185926, Abcam, Cambridge, UK) diluted in blocking buffer, followed by incubation for 45 min at room temperature (RT) with secondary antibodies (anti-rabbit Alexa Fluor® 488: 1:500, cat. #A32790, ThermoFisher Scientific; anti-guinea pig Alexa Fluor® 488: 1:500, cat. #A11073, ThermoFisher Scientific; anti-rabbit Alexa Fluor® 647: 1:500, cat. #A21245, ThermoFisher Scientific) diluted in blocking buffer. Coverslips were then mounted using ProLong antifade (cat. #P36931, ThermoFisher Scientific) and imaged using a Leica SP8 con-focal microscope with a 63X oil-immersion objective.

**Immunohistochemistry.** One month after AAV2/1 injection, mice were deeply anesthetized with an intraperitoneal injection of ketamine/ xylazine (ketamine 100 mg/kg, xylazine 10 mg/kg) and transcardially perfused with ice-cold 0.1 M phosphate buffer (PB; pH 7.4), followed by 4% paraformaldehyde in 0.1 PB. After perfusion, brains were rapidly dissected out and post-fixed in the same fixative solution overnight at 4 °C. After several washes in 0.1 M PB, brains were cryoprotected by immersion in 30% sucrose solutions, subsequently cut in 30 μm sec-tions with a Vibratome and stored at −20 °C in a solution containing 30% ethylene glycol and 20% glycerol in 0.1 M PB. Sections containing the dorsal hippocampi were then washed in PBS and processed for antigen retrieval immunofluorescence. Briefly, pre-washed brain slices were incubated in a sodium citrate buffer (10 mM sodium citrate, 0.05% Tween-20, pH 6.0) and maintained at 95 °C in a heater for 20 min. After 3 washing steps for 2 min, slices were incubated in a blocking solution (10% normal goat serum / 0.1% Triton X-100 in PBS) and then incubated overnight at 4 °C with the following primary anti-bodies: mouse anti-CaMKIIα (1:100, cat. #50049, Cell Signaling, Dan-vers, USA), guinea pig anti-GFP (1:500, cat. #132005, Synaptic Systems GmbH) and rabbit anti-parvalbumin (1:1000, cat. #ab181086, Abcam). Antibodies were diluted in PBS with 3% of normal goat serum and 0.1% Triton X-100. Sections were then washed again in PBS (4 ×10 min) and incubated for 1 h at 25 °C with anti-guinea pig Alexa Fluor 647® (cat. #A-21450, ThermoFisher Scientific) and anti-mouse Alexa Fluor 568® (cat. #A-11004, ThermoFisher Scientific) secondary antibodies diluted 1:500. Nuclear counterstaining was obtained with Hoechst (cat. #H13570, Invitrogen). After several rinses with PBS, sections were mounted on glass slides and observed with a Leica SP8 confocal

microscope (Leica Microsystems). For the analysis of the cell specificity of construct expression, in the dorsal hippocampus, images were acquired with 63x oil-immersion objective using the Navigator Scan mode and slices were reconstructed using the LASX software. The co-localization between the GFP/CaMKII and GFP/PV signals and markers of excitatory and inhibitory neurons was analyzed using ImageJ.

## Protein extraction and western blotting

Tissues, HEK293 cells and hippocampal neurons were lysed using RIPA buffer (0.5 mM EDTA, 20 mM Tris HCl pH 8, 100 mM NaCl, 0.5 % NP-40, 0.5% Na-deoxycholate) supplemented with protease inhibitor cocktail (cat. #04693116001, Roche Applied Science, Penzberg, Germany), phosphatase inhibitor cocktail 2 (cat. #P5726, Sigma-Aldrich, Milano, Italy) and phosphatase inhibitor cocktail 3 (cat. #P0044, Sigma). The final protein concentration was quantified using the BCA protein assay kit (cat. #23225, Pierce Biotech, Waltham, MA). Proteins were separated by sodium dodecyl sulfate polyacrylamide gel electrophoresis (SDS-PAGE) and subsequently transferred to nitrocellulose membranes, following standard protocols. Membranes were incubated in Tris-buffered saline/0.1% Tween-20 (TBST) supplemented with 5% nonfat dry milk (blocking buffer) for 1 h and then incubated with primary antibodies overnight at 4 °C in solution according manufacturer's suggestion. Antibodies to $E^2GFP$ (1:1000, cat. #A111222, ThermoFisher Scientific), *Renilla* luciferase (1:1000, cat. #ab185926, Abcam), α- Actin (1:5000, cat. #A2066, Sigma-Aldrich), calnexin (1:20000, cat. #ADI-SPA-860, Enzo Lifesciences, Farmingdale, NY) and GAPDH (1:10000, cat. #14C10, Cell Signaling). Membranes were then washed several times in TBST incubated with secondary antibodies (goat anti-rabbit IgG (H + L), Peroxidase Conjugated, cat. #PR31460, ThermoFisher Scientific) for 45 min and revealed using the ECL Blotting Reagent (cat. #RPN2109, GE Healthcare). Densitometric analysis was performed with iBright Analysis Software (ThermoFisher Scientific).

## Coelenterazine preparations

Coelenterazine 400a (CTZ 400a) was purchased from Santa Cruz Biotechnology (Dallas, Texas, USA) (cat. #sc-280647) and solubilized in 100% ethanol (1 mM) following manufacturer's recommended protocol. Before in vitro experiments, the stock solution was diluted in 20 mM (2-Hydroxypropyl)-β-cyclodextrin (cat. #H5784, Sigma) to the required concentration (20, 40, 50 µM) and added to the external recording solution. For in vivo experiments, CTZ 400a was diluted in ethanol and brought to a final concentration of 100 µM in saline solution. CTZh (cat. #S2011, Promega) was solubilized in 20 mM (2-Hydroxypropyl)-β-cyclodextrin in PBS as described[73] and used at the final concentration of 20 µM[23]. Frozen stock solutions were kept at −20 °C protected from light.

## In vitro bioluminescence and fluorescence detection

HEK293 cells were plated in a 6-well plate at a concentration of 250,000 cells/well. After 24 h, cells were transfected with pHIL, Ctrl, and RLuc8-S257G alone (pTRE-Tight-RLuc8-S257G, cat. # 79844, Addgene, Watertown, MA). The day after transfection, cells were trypsinized, resuspended and plated in a 96-well plate (cat. #CLS3610, Corning®, Corning, New York, NY) in complete DMEM without phenol red (cat. #31053028, ThermoFisher Scientific) and penicillin/streptomycin at a concentration of 75,000 cell/well. The experiments were performed by adding increasing concentrations of CTZ 400a (10, 20, 50 µM) and recording the resulting luminescence and fluorescence intensities of RLuc8-S257G and $E^2GFP$, respectively using the SPARK 500 F multimode microplate reader (Tecan Group Ltd., Männedorf, Switzerland) in Luminescence Dual Color mode (magenta/green filters) under different pH conditions. Buffers used to change the intracellular pH were composed as follows: pH 6 buffer: 1 mM $MgCl_2$, 10 mM MES, 100 mM KCl, 100 mM NaCl, 1.36 mM $CaCl_2$, 10 mM glucose,

20 µM nigericin (cat. #N7143, Sigma); pH 7.4 buffer: 1 mM $MgCl_2$, 10 mM HEPES, 100 mM KCl, 100 mM NaCl, 1.36 mM, $CaCl_2$, 10 mM glucose, 20 µM nigericin. All experiments were performed at RT in the dark. For the estimation of emission spectra, luminescence was measured using the SPARK 550 F reader (Tecan Group Ltd.) in the luminescence scan mode.

For the live cell detection of bioluminescence, HEK293T cells were plated in a 6-well plate at a concentration of 250,000 cells/well. After 24 h, cells were transfected with pHIL, Ctrl or RLuc8-S257G. The day after transfection, cells were trypsinized, resuspended and plated in a 24-well plate (black 24-well plate with flat and clear bottom, cat. #82426, Ibidi GmbH, Gräfelfing, Germany) in complete DMEM without phenol red and penicillin/streptomycin (cat. #31053028, Thermo-Fisher Scientific) at a concentration of 75,000 cell/well. The experiments were performed by adding CTZ 400a (50 µM) and following the luminescence of RLuc8 and the fluorescence of $E^2GFP$ using a Luminoview bioluminescence imaging system (LV200, Olympus, Tokyo, Japan). The acquisition was carried out using two different filters (Qdot 655 excitation filter and the 510 nm emission filter) under two different pHi conditions. Buffers used to change the intracellular pH were composed as follows: pH 6 buffer: 1 mM $MgCl_2$, 10 mM MES, 100 mM KCl, 100 mM NaCl, 1.36 mM $CaCl_2$, 10 mM glucose, 20 µM nigericin; pH 7.4 buffer: 1 mM $MgCl_2$, 10 mM HEPES, 100 mM KCl, 100 mM NaCl, 1.36 mM, $CaCl_2$, 10 mM glucose, 20 µM nigericin. All experiments were performed at RT in the dark.

## Time-lapse live cell imaging of intracellular pH

Live cell imaging of intracellular pH was performed using a Leica SP8 laser scanning confocal microscope equipped with a 40x oil immersion objective (Leica Microsystems). The sequential excitation of $E^2GFP$ at 405 nm and 488 nm was achieved with a multiline argon laser. Emitted fluorescence was collected between 510 nm and 560 nm using a single photomultiplier tube (PMT) at a constant voltage. For running pH calibration curves, primary hippocampal neurons were plated on poly-D lysine coated 25-mm coverslips, transfected with CD4-$E^2GFP$ at 10 DIV and imaged 48 h later. Coverslips with cells were positioned in a chamber with a K$^+$-enriched buffer composed of: 120 mM potassium gluconate, 40 mM NaCl, 20 mM HEPES, 0.5 mM $CaCl_2$, 0.5 mM $MgSO_4$ pH 7.5. Cells were subsequently perfused with the same buffer, adjusted to the desired pH by adding HEPES for pH>7, NaOH and MES for pH <7, and supplemented with nigericin 10 µM and valinomycin 10 µM. The calibration curve was performed at pH 5.5 to 7.5. Images were taken 10 min after the addition of the buffer supplemented with ionophores[49]. Calibration curves were fitted according to a sigmoidal dose-response curve using the Graphpad Prism 7.04 software. To determine the intracellular pH value during paroxysmal network activity, transfected neurons were first perfused with a buffer composed of 140 mM NaCl, 2.5 mM KCl, 1.8 mM $CaCl_2$, 1 mM $MgCl_2$, 20 mM HEPES, 10 mM glucose, pH 7.4. Baseline fluorescence was collected for 2 min, after which 30 µM bicuculline (BIC; cat. #0131, Tocris, Bristol, UK) was added and fluorescence was collected for additional 8 min. Confocal images were analyzed using the Image J software. For the analysis, regions of interest (ROIs) of ≈ 2 µm² were chosen for each image, positioned on discrete structures visible on neurite extensions. Fluorescence intensity ratios were calculated according to the equation $R = F_{405} / F_{488}$, where F is the average pixel intensity for each ROI when $E^2GFP$ is excited at the respective wavelength[51]. Emission fluorescence ratios were converted to absolute pH values according to the previously obtained calibration curves.

## Patch-clamp recordings

Whole-cell patch-clamp recordings on HEK293 cells and primary hippocampal neurons were performed at room temperature (22-25 °C) for all the experiments. HEK293 cells plated on Petri dishes were transfected with either pHIL or Ctrl plasmids and recorded 3 days later.

Hippocampal neurons were infected with AAVs encoding either pHIL or Ctrl at 3 DIV and recorded at 14-16 DIV.

All recordings were carried out using an EPC10 amplifier and data acquisition was performed using PatchMaster program (HEKA Elektronic, Reutlingen, Germany). During the recordings, cells were maintained in extracellular standard solution (Tyrode) containing: NaCl 140 mM, KCl 4 mM, CaCl$_2$ 2 mM, MgCl$_2$ 1 mM, HEPES 10 mM, Glucose 10 mM, pH 7.3 with NaOH. Patch pipettes, prepared from thin borosilicate glass, were pulled and fire-polished to a final resistance of 3-4 MΩ (HEK293 cells) or 8-10 MΩ (neurons) to avoid total replacement of intracellular environment with the internal standard solution. The intracellular solution was composed by: K$^+$ gluconate 126 mM, NaCl 4 mM, MgSO$_4$ 1 mM, CaCl$_2$ 0.02 mM, BAPTA 0.1 mM, Glucose 15 mM, HEPES 5 mM, ATP 3 mM, GTP 0.1 mM. All the experiments were performed under dark conditions and maintaining CTZ 400a and CTZh at 4 °C in ice to avoid degradation.

For the probe validation in HEK293 cells, the intracellular pH was changed through the patch pipette using an internal solution at three different pH: 6, 7.4 and 8. For the acidic solution (pH 6) HEPES was substituted with MES (5 mM). Recordings were performed in current-clamp configuration at a holding potential of 0 mV. CTZ 400a or CTZh were added directly to the external solution, during the recording, at a final concentration of 10 and 40 µM, respectively.

For the analysis of intrinsic excitability in primary hippocampal neurons, current-clamp recordings were performed at a holding potential of −70 mV, and AP firing was induced by injecting current steps of 10 pA lasting 500 ms. Neurons were maintained in Tyrode solution containing: D-(−)-2-amino-5-phosphonopentanoic acid (D-AP5; 50 µm), 6-cyano-7 nitroquinoxaline-2,3-dione (CNQX; 10 µm), bicuculline methiodide (BIC; 30 µm), and (2 S)−3-[(1 S)−1-(3,4-dichlorophenyl)ethyl] amino-2-hydroxypropyl] (phenylmethyl)phosphinic acid hydrochloride (CGP58845; 5 µm) to block NMDA, non-NMDA, GABA$_A$, and GABA$_B$ receptors, respectively. GABA-dependent membrane potential variations were recorded in current-clamp configuration by maintaining neurons at the resting membrane potential in Tyrode solution, in the absence of synaptic transmission blockers. GABA (100 µM) was added directly in Tyrode solution.

For the probe validation in primary neurons, spontaneous APs were recorded in cell-attached voltage-clamp configuration by clamping neurons in the Heka "on-cell" mode at a holding potential of −70 mV. Neurons were maintained in Tyrode solution in the absence of synaptic transmission blockers and continuously recorded for at least 60 s without applying any current under three distinct external solutions: (i) Tyrode, (ii) Tyrode with either BIC (30 µM) or 4AP (100 µM) and (iii) Tyrode with the convulsant plus either CTZ 400a (40 µM) or vehicle (ethanol 100%/ 2-Hydroxypropyl-β-cyclodextrin mixture). Action potentials and AP bursts were detected using the event detector function of Clampfit 10.7 software (Molecular Devices, Sunnyvale, CA) which identifies as burst a minimum of 4 events within a 200-ms interval.

## MEA extracellular recordings

Neuronal activity was recorded using a multiwell MEA system (Maestro, Axion BioSystems, Atlanta, GA). The MEA plates used (M768-tMEA-48W, Axion BioSystems) contain 48 wells, each with a square grid of 16 electrodes (50 µm electrode diameter; 350 µm center-to-center spacing) that create a 1.1 × 1.1 mm recording area. MEAs, coated by depositing a 20 µl drop of poly-L-lysine (0.1 mg/ml, Sigma-Aldrich) over each recording area, were incubated overnight. Dissociated hippocampal neurons were plated at a final density of 50,000 neurons per well and incubated with Neurobasal medium w/o phenol red supplemented with 1% Glutamax, 2% B27, and 1% penicillin-streptomycin. One-third of the medium was replaced with fresh medium every week. Spiking activity from hippocampal networks grown onto MEAs was monitored and recorded using the Axion BioSystems hardware

(Maestro amplifier and Middle-man data acquisition interface) and the Axion's Integrated Studio software (AxIS 2.4). Neuronal networks on MEAs were infected with AAV2/1 encoding either pHIL or Ctrl at DIV 4. After 10 days, MEA plates were set on the Maestro apparatus and their activity recorded for 10 min at 37 °C (baseline). Networks were then acutely exposed to the GABA$_A$ receptor antagonist, bicuculline (30 µM), and recordings were carried on for further 10 min. The final recordings were made after the addition of either CTZ 400a (40 µM) or vehicle (ethanol 100%/ 2-Hydroxypropyl-β-cyclodextrin mixture). After 1200x amplification, raw data were digitized at 12.5 kHz/channel and stored for subsequent offline analysis. Spike detection and spike train data analysis were computed using the Axion BioSystems software NeuralMetricTool.

## AAV transduction of the dorsal hippocampus

Hippocampal stereotaxic injections (from Bregma: AP −2.0, LAT + /− 1.8; from brain: Z 1.50)[74] were performed in 2-month-old C57Bl6/J male mice using AAV2/1 encoding either CaMKIIα-pHIL or CaMKIIα-Ctrl. Anesthesia was induced by exposure to 1-4% isoflurane. Mice were placed in a stereotaxic frame and the head adjusted to a flat-skull position. A small craniotomy was performed bilaterally at the injection coordinates indicated above and AAV1/2 particles were injected in the hippocampus via a glass pipette (1 µL/site at a flow rate of 0.1 µL/min). The injection pipette was left in place for at least 5 min at the end of each injection to allow the complete diffusion of the virus. After injection, mice were returned to their home cage and administered with ketoprofen 0.5-1 mg/kg, dexamethasone 5 mg/kg to speed up recovery from anesthesia. After brain surgery, animals were single housed with standard environmental enrichment (nest material and plastic house) and *ad libitum* access to food and water and allowed to recover for at least 4 weeks before behavioral experiments.

## In vivo bioluminescence detection

One month after AAV injections, mice were shaved on the head and injected in the tail vein with CTZ 400a (0.15-0.6 mg/kg). Immediately after, anesthesia was induced by exposure to 1%-4% isoflurane and animals were monitored using the IVIS imaging apparatus (Lumina II, Perkin Elmer, Milan, Italy) at various time intervals (every 5 min for 20 min) to study the time-course of endogenous bioluminescence. The bioluminescence emitted from the hippocampi in both hemispheres were recorded, analyzed based on the radiance detected in the ROIs and expressed in p/sec/cm$^2$/sr. After the live imaging and a recovery period of 2-7 days, mice were used for behavioral experiments.

## Behavioral experiments

A battery of behavioral tests was performed 4-5 weeks after brain surgery, to evaluate the general locomotor activity levels, anxiety and hippocampal functions after the AAV transduction of the hippocampus. Behavioral experiments were conducted during the standard light phase (between 10.00 a.m. and 6 p.m.) in the following order: open field, novel object recognition and contextual fear conditioning tasks. Animals were habituated to the experimental room for one hour before each behavioral task.

**Open field.** To evaluate both locomotor activity and anxiety-like behaviors, each subject underwent an open field test. Mice were individually placed in an opaque open field box (40x40x40 cm) with an Anymaze camera (Stoelting, Wood Dale, IL) mounted 20 cm above the apparatus. The session started by placing the animal in the center of the arena for 10 min. The floor of the apparatus was cleaned with 50% ethanol after each test. Total distance, time locomotion, time immobile, mean speed, as well as time in the center and in the border were automatically collected using the Anymaze activity monitor and analyzer software system (Supplementary Fig. 9A).

**Novel object recognition.** The novel object recognition test was conducted in the open field arena. The test consisted of 10 min habituation to the arena, 10-min familiarization session with two identical objects and a 10 min recognition of a novel object. During the familiarization session, two identical objects were placed in the arena. Two hours after the familiarization session, a 10-min novel recognition test started. One clean familiar object and one clean novel object were placed in the arena. The familiarization sessions and the recognition test were videotaped and manually scored. Object investigation was defined as the time spent in contact with the object. Recognition memory was defined as spending significantly more time sniffing the novel object than the familiar object (Supplementary Fig. 9B).

**Contextual fear conditioning.** The contextual fear-conditioning test was conducted in the Video Fear Conditioning apparatus (Med Associates Inc., Fairfax, VT). The chamber consisted of a grid floor connected with a circuit board for the delivery of a mild electric shock. A camera mounted on the front door of the chamber recorded test sessions, which were automatically scored using the Med. Associates software. During the conditioning phase (day1), the mouse was placed in the test chamber, allowed to explore freely for 2 min, and immediately after subjected to three footshocks (2 s, 0.5 mA) with an inter-trial intervals of 60 s. After the last electric shock, the mouse was left in the chamber for another 2 min, during which freezing behavior was scored. Contextual fear memory was tested 24 h later in the same chamber, but without foot shock. Each mouse was placed in the chamber for 5 min and the freezing behavior was scored. After one hour, animals were retested in a novel context. The chamber was altered by covering the grid floor with a smooth white plastic sheet and by replacing the black arena with black and white stripped walls. Animals were placed in the novel context for 5 min. and freezing behavior was scored. The chamber was thoroughly cleaned of odors between sessions, using 50% ethanol and water (Supplementary Fig. 9C).

The same behavioral tests were also performed in transduced mice after the administration of either CTZ 400a (0.3 and 0.6 mg/kg) or the related vehicle (10% and 20% ethanol in 150 μl saline, respectively). In the novel object recognition test, CTZ 400a was alternatively administered before familiarization or before recognition (Supplementary Fig. 10B), while in the contextual fear conditioning test CTZ 400a was alternatively administered before the training session or before the fear context session (Supplementary Fig. 10C).

**Phenotypic analysis of pilocarpine-induced and sound-induced seizures in wild type and PRRT2 KO mice**
**Pilocarpine induced seizures in wild type mice.** One week after the evaluation of bioluminescence in vivo, mice were subjected to a single dose of pilocarpine to evaluate the anti-seizure activity of pHIL. CTZ 400a was injected in the tail vein (0.15-0.6 mg/kg in 150 μl) and immediately after, an intraperitoneal administration of pilocarpine (300 mg/kg, cat. #141887, Sigma Aldrich) or vehicle was performed. Seizure score was adapted from a previously documented scale[75] with modifications to make it suitable for the pilocarpine-triggered seizure phenotype: 0 = immobility, 2 = head and body twitches, 4 = forelimb clonus (forepaws begin to shake), 4 = Straub tail with or without a shake, 5 = full tonic-clonic seizures. Mice were recorded over 1 h and analyzed by inspection of the videos to classify the behavioral seizures.

**Sound induced seizures in PRRT2 KO mice.** To evaluate the anti-seizure effect of pHIL in PRRT2 KO mice, we induced audiogenic seizures as previously described[56]. The test was conducted in a circular apparatus (15-cm diameter) located inside a Startle response/PPI test system chamber (TSE Systems GmbH, Germany). The session consisted of a 2-min free exploration period, followed by 1-min of acoustic stimulation (white noise at 120 dB) and 2 more min of free exploration.

All sessions were recorded with a digital camera (HD Professional Webcam C920; Logitech, Lausanne, Switzerland) placed above the experimental chamber. Audiogenic responses were scored manually as: wild running, tonic-clonic seizures and post-ictal immobility. The test was conducted between 03:00 p.m. and 05:00 p.m. to avoid possible circadian variations. All animals were first tested without CTZ administration. After one week, the behavioral test was repeated in the same mice injected in the tail vein with CTZ 400a (0.3 mg/kg in 150 μl) right before the task. Jump duration was analyzed by using a classical startle apparatus. The test was conducted one week after the first audiogenic task and was repeated before and after CTZ400a administration.

## Statistics
The number of samples necessary for the experiments was preliminarily calculated based on the experimental variability and need to reach an appropriate number of replications for a robust statistical analysis. The number of animals/independent cell preparations for the planned experiments (sample size, n) was predetermined using the G*Power software according to the following formula: $n = Z^2 \times \sigma^2 / \Delta^2$, where: Z is the value of the distribution function f(α,β) (with α and β type-I and type-II errors, respectively; based on α = 0.05 and 1-β = 0.9), σ is the standard deviation of the groups (set between 0.2-0.3 based on similar experiments and preliminary data) and Δ the minimum percent difference that is thought to be biologically relevant (0.2 or 20%). Experiments were carried out blind to the experimenter. Data are expressed either as means ± SEM for number of independent animals (n) with superimposition of the individual experimental points or as box plots characterized by the following elements: center line, median (Q2); cross symbol, mean; box limits, 25th (Q1)−75th (Q3) percentiles; whisker length refers to min-to max values. Normal distribution was assessed using the D'Agostino−Pearson normality test. To compare two normally distributed sample groups, the unpaired or paired two-tailed Student's t-test was used. To compare two sample groups that were not normally distributed, the Mann-Whitney's U-test was used. To compare more than two normally distributed sample groups, one- or two-way ANOVA followed by post-hoc multiple comparisons (Holm-Šidák's, Bonferroni's or Tukey's tests). To compare more than two normally distributed sample groups, one- or two-way ANOVA followed by post-hoc multiple comparisons (Holm-Šidák's, Bonferroni's or Tukey's tests). To compare more than two non-normally distributed sample groups, we used either the Kuscal-Wallis one-way ANOVA or the paired Friedman's test followed by the post-hoc Dunn's test. Contingency analysis was carried out by using the Fisher's exact test. $p < 0.05$ was considered significant. Statistical analysis was carried out using Prism v9 (GraphPad Software). The representative images not supported by the relevant statistics have been chosen upon 3 independent preparations with similar outcome.

## Reporting summary
Further information on research design is available in the Nature Portfolio Reporting Summary linked to this article.

# Data availability
All data supporting the findings of this study are available within the article and its supplementary files. Any additional requests for information can be directed to and will be fulfilled by the corresponding authors. Source data are provided with this paper.

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

## Acknowledgements

We thank Drs. Ranieri Bizzarri (Institute of Biophysics, Italian National Research Council, Pisa, Italy) for kindly providing the E2GFP construct, Michele Cilli and Sebastian Sulis Sato (IRCCS Ospedale Policlinico San Martino, Genova, Italy) for useful suggestions for the in vivo experiments. We also thank R. Ciancio, I. Dallorto, A. Mehilli, R. Navone and D. Moruzzo (Istituto Italiano di Tecnologia, Genova, Italy) for technical and administrative assistance. The study was supported by grants from Euronanomed3 (project Nanolight 2019-132 to FB), the Italian Ministry of University and Research (PRIN 2020WMSNBL to FB; PRIN-Next Generation EU P2022EZ9LN to FB and FC), the Italian Ministry of Health (Next Generation EU *PNRR-MR1-2022-12376528* to FB) and IRCCS Ospedale Policlinico San Martino Genova (Ricerca Corrente and 5x1000 grants to FB and EC).

## Author contributions

A.M. engineered and characterized the constructs; MM and AR performed the in vitro electrophysiological experiments; A.M., L.C., and F.V. carried out the surgical procedures and performed the in vitro and in vivo expression studies; C.M., L.E., and S.A. performed the in vivo bioluminescence characterization; SH performed the bioluminescence studies in vitro; C.M. and L.C. performed the behavioral studies; K.B. and R.E.G. provided the ILMO2 construct and revised the manuscript; F.C., E.C., and F.B. conceived and supervised the study and analyzed the data; F.B. and E.C. wrote the paper; FB financially supported the study; all authors critically discussed the experimental results and contributed to manuscript writing and revision.

## Competing interests

The authors declare no competing interests.
