## [Peer Review File · Nature Communications]

REVIEWER COMMENTS

Reviewer #1 (Remarks to the Author):

In this study, the authors designed and implemented a closed-loop chemo-optogenetic nanomachine named pH-sensitive inhibitory luminopsin (pHIL), composed of a luciferase-based light generator, a fluorescent sensor of intracellular pH (E2GFP), and an optogenetic actuator (halorhodopsin) for silencing neuronal activity. The authors first showed that E2GFP is an excellent sensor of intracellular acidification in both human embryonic kidney 293T (HEK 293) cells and hippocampal neurons. Next, the authors designed and implemented pHIL as a triple chimera composed of RLuc8, E2GFP, and NpHR. The construct is fueled by endogenous RLuc8-generated light (BRET) and FRET to activate the inhibitory opsin NpHR. After the luciferase substrate (CTZ 400a) is administered to switch on the BRET from RLuc8, the cascading FRET in the pHIL makes it responsive to low intracellular pH and optogenetically hyperpolarizes the cells. To demonstrate this closed-loop nanomachine's efficacy in aborting hypersynchronous activities in neurons, the authors transduced primary hippocampal neurons with pHIL or control construct (without NpHR). In the presence of 4AP and CTZ400a, the pHIL could suppress 4-AP-induced neuronal hyperactivity, whereas the control construct did not. pHIL also effectively suppresses neuronal hyperactivity induced by GABAA receptor blockade from bicuculline (BIC) in vitro. Finally, mice dorsal hippocampal neurons were transduced with either pHIL or control construct using AAV injection under CAMKIIa promoter. The mice were then treated with IP injection of pilocarpine followed by injection of CTZ400a. One hour after the chemical administration, there was no difference in the behavioral manifestation of milder seizures between animals with pHIL and those with control construct, but there was a longer delay in the behavioral manifestation of more severe seizures in animals with pHIL than those with control construct. The animals with pHIL also had fewer and shorter general tonic-clonic seizures than those with the control construct.

Major concerns:

1. The authors provided strong evidence that E2GFP is a suitable sensor for intracellular pH levels and some evidence that hypersynchronous activities are associated with lower intracellular pH. There are potential pitfalls with this approach.

First, since pHIL acts at a single cell level, the inhibitory drive may impact normal cognitive functions. The authors provided evidence that animals with pHIL without the influence of CTZ do not show any significant cognitive impairment in the open field, novel object, and contextual fearing condition tests (supple. Fig 6A-C). The cognitive performances of animals with pHIL should also be evaluated with the animals under the influence of CTZ. This is necessary because the animals' cognition and behavior may be impacted by the closed-loop inhibition of the glutamergic neurons

in the hippocampus if there is a false positive activation of NpHR. The effect of pHIL on an animal's cognition may depend on the dose of CTZ. There is an optimal range of doses for CTZ to be effective in stopping seizures, but it has little or no impact on cognition. This would require a dose-response assessment.

Second, the change in intracellular pH may result from other physiological or pathological events, such as physical altitude, trauma, or stroke. The effect of the pHIL on these processes is unknown. It would be important for the authors to discuss the ramifications of using intracellular pH as a biomarker for seizure onset.

2. The control construct lacks a transmembrane component and has a different distribution in cellular compartments from pHIL. This may impact its effect on cellular functions in the presence of lower intracellular pH and CTZ400a. A control construct with a functionally silent transmembrane component may be a more suitable choice. For example, can the investigator use a control construct with a transmembrane component similar to the one they used to test whether the E2GFP is a suitable sensor of intracellular acidification where a CD4 receptor was used to anchor the E2GFP to the cell membrane?

3. The in-vivo pilocarpine results demonstrate only minor effects and are not very convincing. The experiment may need more thought. First of all, the timeline of the pHIL activation decays very quickly, as shown in Fig 7A and B. By the time the IP pilocarpine had a substantial effect (~10-20 min after injection), pHIL is largely inactive. This casts doubt on whether the results observed are indeed from pHIL activation. Have the authors tried changing the order of injections, injecting CTZ400a ~10 mins after pilocarpine? Have the authors tried higher dosages of CTZ400a or methods to achieve more lasting activation of pHIL?

Secondly, the data points for tonic-clonic seizures in 7D, E, and F do not seem to match up. Can the authors elaborate on whether they are from the same cohort of mice and the total number of mice used in these experiments? In 7E, what does it mean for the data points with 60 60-minute duration? Were these data points from the Ctrl group observed with 0min latency for tonic-clonic in 7D? If so, those data points are missing from the plot.

Also, in 7E, what does it mean for the data points 0 duration? Do they correspond to mice without tonic-clonic seizures? If so, this appears to be 2/28 and 6/29 for Ctrl and pHIL, respectively, then 6/29 ~80% does not match what's plotted in 7F. In any case, I could not reproduce the p-value that the authors claimed for 7F using Fisher's exact test unless 3x the animal count is used.

4. There is likely a dose-dependent response to CTZ400a in animals with pHIL construct regarding its efficacy in reducing seizure frequency and severity after administering pilocarpine. I recommend that the investigators include this assessment.

Minor concerns:

1. It's unclear why the investigation of different gene expression levels was conducted on HEK293 cells and hippocampal neurons.
2. BRET and FRET were introduced for the first time in the result section on page 7. They should be spelled out.
3. In Figure 2E, I recommend using different color choices for Ctrl and RLuc8 to make them more distinct.
4. In Figure 6B, the authors could reduce potential confusion if the bar plot categories Cer, Cx, and H are presented in the same order as the Westin plot results on the left.
5. I recommend using modified Racine scores to characterize seizure phenotypes in the in vivo pilocarpine experiments.

Reviewer #2 (Remarks to the Author):

This manuscript presents the development and validation of a closed-loop chemo-optogenetic nanomachine, pHIL, as a novel gene therapy approach for the treatment of epilepsy. This tool works through two steps: (1) a bioluminescence resonance energy transfer (BRET) from a luciferase-based light generator (RLuc8) to a sensor of intracellular pH (E2GFP) and (2) a fluorescence resonance energy transfer (FRET) from E2GFP to halorhodopsin. This latter transfer occurs specifically in the context of intracellular acidification, which results from neuronal hyperactivity. Therefore, pHIL is a closed-loop system in which neuronal hyperactivity triggers neuronal inhibition at the level of the individual neuron.

This is an exciting and highly innovative approach that may offer significant advantages relative to traditional closed-loop optogenetic stimulation, in which hyperactivity measured at the macro-scale – i.e., seizure onset detected via EEG – triggers an exogenous light source, which then inhibits all opsin-expressing neurons in the region. In contrast, pHIL can act at the single neuron level (which could theoretically work faster and with fewer off-target effects), is not spatially constrained by exogenous light delivery (so may be more effective in the case of large or multifocal seizure onset zones) and does not require implanted hardware. However – and especially in light of other

recently-developed biochemical closed-loop strategies (using extracellular glutamate concentrations or cFOS as hyperexcitability sensors) – several key aspects of the pHIL approach require clarification in order to fully assess how it fits into the existing landscape, and its potential impact as a tool for the epilepsy field.

Major concerns:

1. Timing is everything for anti-seizure tools. The data included in the manuscript do not provide sufficient clarity regarding the activation kinetics of pHIL. At a single neuron level, what is the temporal lag from the onset of neuronal hyperexcitability to the activation of halorhodopsin? (The data in figures 4, 5, and 7 do not answer this question.) Traditional closed-loop optogenetics takes effect (on a macro-scale) on a timescale of several seconds (limited by EEG seizure detection). How does pHIL compare?

2. Related to the above, many spontaneous seizures self-terminate rapidly (< 2 minutes) so an inhibitory tool will presumably need to activate very rapidly at seizure onset to make any meaningful difference in seizure duration. Is pHIL effective at terminating spontaneous seizures? Systemic pilocarpine injection evokes prolonged ictal activity (i.e., status epilepticus) so pHIL may show an effect even if its onset takes many seconds or even minutes. The impact of this manuscript would be greatly enhanced by the inclusion of data from a spontaneous seizure model (as in both Qui et al 2022 and Lieb et al 2018). Note that convincing data showing efficacy aborting spontaneous seizures in vivo would reduce the burden of proof regarding precise kinetics within individual neurons.

3. pHIL functions only in the presence of an exogenous luciferase substrate (CTZ). To understand to what extent this is a burdensome requirement, more information is needed regarding the in vivo kinetics of CTZ. Do the results from Figure 7A-B suggest that pHIL will only work for < 15 minutes following systemic CTZ injection? That seems to be a prohibitive constraint, assuming a goal of inhibiting spontaneous seizures! It would be helpful for the authors to clarify the time window available for experimentation following CTZ injection, either with additional experiments, if necessary, or with modifications to the text if no further data is required to adequately answer this question. The authors should explicitly address this experimental constraint – and future options (if any) to attenuate it – in the discussion.

4. It would be helpful to draw some practical and/or conceptual distinctions between this novel pHIL strategy and the other closed-loop “hyperactivity sensor” approaches. The cFOS strategy is fundamentally different in terms of timing: the Kv1.1 expression occurs too late to abort the first, inciting seizure but sticks around long enough to inhibit subsequent seizures occurring within a timescale of ~48 hours; it would be helpful for the reader if the authors made this conceptual

distinction clearer in the text. On the other hand, the extracellular glutamate strategy seems more conceptually similar to pHIL... What would be advantages or disadvantages of one or another approach?

Minor concerns:

1. Lines 65-68 / lines 406-407: I agree with the authors' intended point that seizures may start within one focus and subsequently spread. However, the specific epilepsy etiologies referenced here do not necessarily correspond to focal onset seizures. This point may be better served with reference to epilepsies arising from focal cortical dysplasia, traumatic brain injury, stroke, etc.

2. Line 126 & 148: The use of the word "precocious" is confusing to me. Do the authors mean that pH changes occur rapidly? (How rapidly?)

3. The data presented in Figure 6 could potentially be moved to the supplement. The authors' careful validation experiments are appreciated but has already been established that the CAMKIIa promoter can be used to achieve expression in excitatory neurons.

4. Line 398, 409, etc.: I suggest the terminology "anti-seizure" rather than "anti-epileptic", since this strategy targets seizures rather than mechanisms of epilepsy per se.

ANSWERS TO REVIEWERS' COMMENTS

Reviewer #1

Major concerns:

1. The authors provided strong evidence that E2GFP is a suitable sensor for intracellular pH levels and some evidence that hypersynchronous activities are associated with lower intracellular pH. There are potential pitfalls with this approach.

First, since pHIL acts at a single cell level, the inhibitory drive may impact normal cognitive functions. The authors provided evidence that animals with pHIL without the influence of CTZ do not show any significant cognitive impairment in the open field, novel object, and contextual fearing condition tests (supple. Fig 6A-C). The cognitive performances of animals with pHIL should also be evaluated with the animals under the influence of CTZ. This is necessary because the animals' cognition and behavior may be impacted by the closed-loop inhibition of the glutamergic neurons in the hippocampus if there is a false positive activation of NpHR. The effect of pHIL on an animal's cognition may depend on the dose of CTZ. There is an optimal range of doses for CTZ to be effective in stopping seizures, but it has little or no impact on cognition. This would require a dose-response assessment.

We thank the Reviewer for the positive evaluation of our strategy and for pointing out potential pitfalls. The fact that pHIL works as a hybrid homeostatic mechanism at the single cell level can make it very effective when activated. On the other hand, activation only occurs in the presence of intracellular acidosis which is not occurring during normal behavioral activity.

However, we totally agree with the Reviewer's suggestions and have run the required experiments. Particularly, we have investigated the cognitive performances of animals expressing pHIL in the excitatory neurons of the hippocampus under the effect of CTZ. Moreover, we performed a CTZ dose-response to identify a dose effective on ameliorating the seizure manifestations, but not affecting the behavioral phenotype. We found that a 50% CTZ dose abolished the effect on seizures, while doubling the dose did not significantly improve the anti-seizure activity. None of the CTZ doses tested altered behavioral performances in both open field, NOR and contextual conditioning. The results of these new experiments have been added to the revised version of the manuscript as new main **Figure 6** and new **Supplementary Figure 10**.

Second, the change in intracellular pH may result from other physiological or pathological events, such as physical altitude, trauma, or stroke. The effect of the pHIL on these processes is unknown. It would be important for the authors to discuss the ramifications of using intracellular pH as a biomarker for seizure onset.

We agree that intracellular acidosis could result from a series of pathological conditions including brain trauma and stroke. Under these conditions, acidosis is believed to reflect pathological hyperactivity of neurons consequent to an excessive release of glutamate. Thus, shutting down neuronal activity under these conditions could be beneficial. We are not positive that physical altitude causes acidosis, as it induces hypocapnia secondary to hyperventilation and thereby extracellular alkalosis. We now deal with these interesting ramifications in the Discussion section (pages 14-15).

2. The control construct lacks a transmembrane component and has a different distribution in cellular compartments from pHIL. This may impact its effect on cellular functions in the presence of lower intracellular pH and CTZ400a. A control construct with a functionally silent transmembrane component may be a more suitable choice. For example, can the investigator use a control construct with a transmembrane component similar to the one they used to test whether the E2GFP is a suitable sensor of intracellular acidification where a CD4 receptor was used to anchor the E2GFP to the cell membrane?

The control construct had the role of expressing a silent protein in the very same neurons activated by CTZ but devoid of a membrane effector. We still believe it is an excellent control for assessing the effects of pHIL. However, as the Reviewer suggested, we engineered a functionally silent transmembrane control construct in which RLuc and E2GFP were brought to the membrane by

CD4. The results, depicted in the new **Supplementary Figure 5**, show that activation of such a construct with CTZ does not affect the hypersynchronous activity induced by 4AP, as observed with our cytosolic control construct.

3. The in-vivo pilocarpine results demonstrate only minor effects and are not very convincing. The experiment may need more thought. First of all, the timeline of the pHIL activation decays very quickly, as shown in Fig 7A and B. By the time the IP pilocarpine had a substantial effect (~10-20 min after injection), pHIL is largely inactive. This casts doubt on whether the results observed are indeed from pHIL activation. Have the authors tried changing the order of injections, injecting CTZ400a ~10 mins after pilocarpine? Have the authors tried higher dosages of CTZ400a or methods to achieve more lasting activation of pHIL?

We did not change the order of CTZ and pilocarpine injections, as the role of pHIL should be to prevent/ameliorate seizure manifestations. In addition, an intravenous injection under pilocarpine-induced tail rigidity is unfeasible. As mentioned above, we have tried a higher dose with respect to the used one, yielding a larger and more long-lasting generation of endogenous light (main new **Figure 6**), but with similar results on the seizure manifestations. The intravenous administration of CTZ is, at the current stage, a proof-of-concept for our strategy and will require future optimization and testing alternative administration routes.

Secondly, the data points for tonic-clonic seizures in 7D, E, and F do not seem to match up. Can the authors elaborate on whether they are from the same cohort of mice and the total number of mice used in these experiments? In 7E, what does it mean for the data points with 60 60-minute duration? Were these data points from the Ctrl group observed with 0min latency for tonic-clonic in 7D? If so, those data points are missing from the plot.

Data in the former Figure 7D-F (now Figure 7B-D) are from the same cohort of mice. In former 7E (now 7C), the maximum duration of the tonic-clonic seizure was 60 sec, not min. This was the longer time of seizure analysis, as correctly mentioned in the Materials and Methods. Just to clarify, Figure 7D (now 7B) refers to latency to seizure, while former Figure 7E (now 7C) refers to duration of tonic-clonic seizures for the animals that reached that stage. There is no direct relationship between latency to, and duration of, tonic-clonic seizures.

Also, in 7E, what does it mean for the data points 0 duration? Do they correspond to mice without tonic-clonic seizures? If so, this appears to be 2/28 and 6/29 for Ctrl and pHIL, respectively, then 6/29 ~80% does not match what's plotted in 7F. In any case, I could not reproduce the p-value that the authors claimed for 7F using Fisher's exact test unless 3x the animal count is used.

Data points at "0 duration" in the former Figure 7E (now 7C) mean that those mice did not experience a tonic-clonic seizure. As in the former Figure 7E (now 7C), these tonic-clonic seizure-free mice were 2/27 and 7/29, as correctly reported in the legend and in the plot as individual points. Thus, the percentages of tonic-clonic seizure-free mice shown in the former Fig. 7F (now 7D) are indeed correct. However, the Reviewer is right in noticing that the Fisher's test significances were incorrect, because the data were erroneously entered as percentages instead of number of animals. We now removed the significance and replaced panel F with pie charts.

4. There is likely a dose-dependent response to CTZ400a in animals with pHIL construct regarding its efficacy in reducing seizure frequency and severity after administering pilocarpine. I recommend that the investigators include this assessment.

As mentioned above, we performed a CTZ dose-response to identify a dose effective on ameliorating the seizure manifestations, but not affecting the behavioral phenotype. We found that a 50% CTZ dose abolished the effect on seizures, while doubling the dose did not significantly improve the anti-seizure activity.

Minor concerns:

1. It's unclear why the investigation of different gene expression levels was conducted on HEK293 cells and hippocampal neurons.

HEK293 represent a much easier system to analytically evaluate BRET/FRET using spectrophotometry and are much more robust cells allowing the test changes of pH using ionophores.

2. BRET and FRET were introduced for the first time in the result section on page 7. They should be spelled out.

Done, thank you.

3. In Figure 2E, I recommend using different color choices for Ctrl and Rluc8 to make them more distinct.

Done.

4. In Figure 6B, the authors could reduce potential confusion if the bar plot categories Cer, Cx, and H are presented in the same order as the Westin plot results on the left.

Fully agree. We have done it, but the figure is now new **Supplementary Figure 8**, as suggested by another Reviewer.

5. I recommend using modified Racine scores to characterize seizure phenotypes in the in vivo pilocarpine experiments.

Indeed, we have used a modified Racine seizure score in which the originally reported scale was adapted to the seizure phenotype triggered by pilocarpine. We better clarify this point in the Methods.

Reviewer #2

This is an exciting and highly innovative approach that may offer significant advantages relative to traditional closed-loop optogenetic stimulation, in which hyperactivity measured at the macro-scale – i.e., seizure onset detected via EEG – triggers an exogenous light source, which then inhibits all opsin-expressing neurons in the region. In contrast, pHIL can act at the single neuron level (which could theoretically work faster and with fewer off-target effects), is not spatially constrained by exogenous light delivery (so may be more effective in the case of large or multifocal seizure onset zones) and does not require implanted hardware. However – and especially in light of other recently-developed biochemical closed-loop strategies (using extracellular glutamate concentrations or cFOS as hyperexcitability sensors) – several key aspects of the pHIL approach require clarification in order to fully assess how it fits into the existing landscape, and its potential impact as a tool for the epilepsy field.

We thank the Reviewer for the positive evaluation of our work.

Major concerns:

1. Timing is everything for anti-seizure tools. The data included in the manuscript do not provide sufficient clarity regarding the activation kinetics of pHIL. At a single neuron level, what is the temporal lag from the onset of neuronal hyperexcitability to the activation of halorhodopsin? (The data in figures 4, 5, and 7 do not answer this question.) Traditional closed loop optogenetics takes effect (on a macro-scale) on a timescale of several seconds (limited by EEG seizure detection). How does pHIL compare?

We fully agree with the Reviewer. We recorded the temporal lag from the activation of the closed loop molecular device by CTZ administration to the activation of halorhodopsin at the single neuron level. The data are shown in the new **Supplementary Figure 4**. The effect is very fast and becomes significant with a 2-s delay.

2. Related to the above, many spontaneous seizures self-terminate rapidly (< 2 minutes) so an inhibitory tool will presumably need to activate very rapidly at seizure onset to make any meaningful difference in seizure duration. Is pHIL effective at terminating spontaneous seizures? Systemic pilocarpine injection evokes prolonged ictal activity (i.e., status epilepticus) so pHIL may show an effect even if its onset takes many seconds or even minutes. The impact of this manuscript would be greatly enhanced by the inclusion of data from a spontaneous seizure model (as in both Qui et al 2022 and Lieb et al 2018). Note that convincing data showing efficacy aborting spontaneous seizures in vivo would reduce the burden of proof regarding precise kinetics within individual neurons.

The efficacy of pHIL in terminating or suppressing spontaneous seizures is a fundamental topic. Based on the proof-of-concept of pHIL activity demonstrated in this paper, we are currently investigating the effects of this inhibitory tool on spontaneous seizures that occur in chronic epilepsy. As spontaneous seizures occur unpredictably and are often very brief, as the Reviewer correctly points out, the application of pHIL to chronic epilepsy requires an optimization of the administration route of CTZ, that we are currently carrying out. With respect to the two excellent papers quoted by the Reviewer, pHIL needs the administration of the luciferase substrate, which makes the issue of timing in the administration critical. Nevertheless, we have included in the new main **Figure 8** the evaluation of the efficacy of pHIL in counteracting epileptic seizures in a genetic model of experimental epilepsy, the PRRT2 knockout mouse. This mouse does not experience spontaneous seizures but has a low seizure threshold so that seizures are easily triggered by environmental stimuli such as sound (Michetti et al., *Neurobiol. Dis.*, 2017). The results clearly demonstrate that pHIL is highly effective in suppressing or greatly weakening the epileptic paroxysm in adult PRRT2 knockout mice.

3. pHIL functions only in the presence of an exogenous luciferase substrate (CTZ). To understand to what extent this is a burdensome requirement, more information is needed regarding the in vivo kinetics of CTZ. Do the results from Figure 7A-B suggest that pHIL will only work for < 15 minutes following systemic CTZ injection? That seems to be a prohibitive constraint, assuming a goal of

inhibiting spontaneous seizures! It would be helpful for the authors to clarify the time window available for experimentation following CTZ injection, either with additional experiments, if necessary, or with modifications to the text if no further data is required to adequately answer this question. The authors should explicitly address this experimental constraint – and future options (if any) to attenuate it – in the discussion.

Looking at the light detected by in vivo luminescence imaging, the Reviewer is right in pointing out the limited time interval (~20 min) of endogenous light generation following CTZ administration. However, the visible light generated inside the hippocampus from both luciferase and GFP must cross the overlying cortex, skull and mouse skin to generate a detectable signal in the external imaging system. Moreover, the light detected upon CTZ administration is the complementary effect of the functional non-radiative energy transfer occurring at molecular level, which indeed might be still in place although not detectable by a whole body luminescence scan equipment. Thus, it is likely that fully effective BRET/FRET mechanisms are generated, at the single cellular level, for much longer times. Said that, we performed, as suggested, a CTZ dose-response to identify a dose effective on ameliorating the seizure manifestations, but not affecting the behavioral phenotype. We found that doubling the CTZ dose (0.6 mg/kg) did not significantly improve the anti-seizure activity with respect to 0.3 mg/kg when CTZ was administered 10 min before pilocarpine. However, it significantly increased the intensity of the generated light, prolonging the time window for an effective activation of the pHIL sensor-actuator. See new main **Figure 6** (panels D-F), new main **Figure 7** (panel E), and new **Supplementary Figure 10**.

4. It would be helpful to draw some practical and/or conceptual distinctions between this novel pHIL strategy and the other closed loop “hyperactivity sensor” approaches. The cFOS strategy is fundamentally different in terms of timing: the Kv1.1 expression occurs too late to abort the first, inciting seizure but sticks around long enough to inhibit subsequent seizures occurring within a timescale of ~48 hours; it would be helpful for the reader if the authors made this conceptual distinction clearer in the text. On the other hand, the extracellular glutamate strategy seems more conceptually similar to pHIL. What would be advantages or disadvantages of one or another approach?

It is certainly important, as the Reviewer suggests, to expand the comparison between the closed loop “hyperactivity sensor” approaches existing to date, mostly the cFos and eGluCl strategies in terms of pros and cons. We now dedicate a full paragraph of the Discussion section to this very important issue (page 16).

Minor concerns:

1. Lines 65-68 / lines 406-407: I agree with the authors’ intended point that seizures may start within one focus and subsequently spread. However, the specific epilepsy etiologies referenced here do not necessarily correspond to focal onset seizures. This point may be better served with reference to epilepsies arising from focal cortical dysplasia, traumatic brain injury, stroke, etc. Thank you for the suggestion. We more precisely refer to typical focal onset epilepsies in the Introduction and Discussion.

2. Line 126 & 148: The use of the word “precocious” is confusing to me. Do the authors mean that pH changes occur rapidly? (How rapidly?)
Corrected.

3. The data presented in Figure 6 could potentially be moved to the supplement. The authors’ careful validation experiments are appreciated but has already been established that the CAMKIIa promoter can be used to achieve expression in excitatory neurons.
Excellent suggestion. The former Figure 6 has now been moved to the Supplementary Information as new **Supplementary Figure 8**.

4. Line 398, 409, etc.: I suggest the terminology “anti-seizure” rather than “anti-epileptic”, since this strategy targets seizures rather than mechanisms of epilepsy per se.
Absolutely right. We corrected both sentences.

REVIEWERS' COMMENTS

Reviewer #1 (Remarks to the Author):

The authors have fully addressed all the concerns and comments I have raised in my review of their manuscript in the revised manuscript.

Their responses were thoughtful and thorough and included additional experiments and data analyses.

I endorse the publication of their manuscript.

Reviewer #2 (Remarks to the Author):

This manuscript presents the development and validation of a closed-loop chemo-optogenetic nanomachine, pHIL, in which neuronal hyperactivity triggers neuronal inhibition at the level of the individual neuron. This is an exciting and innovative novel gene therapy approach for the treatment of epilepsy.

The revised manuscript is significantly strengthened by a large quantity of additional control and characterization data, addition of a second mouse epilepsy model, and expansion of the discussion section. In my opinion this manuscript will be of great interest to the scientific community.